# LAUGHING HYENA DISTILLERY
## Extracting Compact Recurrences From Convolutions

**Stefano Massaroli**[*,1], **Michael Poli**[*,2], **Daniel Y. Fu**[*,2],
**Hermann Kumbong**[2], **Rom N. Parnichkun**[3], **Aman Timalsina**[4],
**David W. Romero**[5], **Quinn McIntyre**[2], **Beidi Chen**[6], **Atri Rudra**[7], **Ce Zhang**[8],
**Christopher Re**[2,†], **Stefano Ermon**[2,†], **Yoshua Bengio**[1,†]

[*]Equal contribution. † Equal senior authorship. [1]Mila and Université de Montréal. [2]Stanford University.
[3]The University of Tokyo. [4]Purdue University. [5]Vrije Universiteit Amsterdam. [6]Carnegie Mellon University
and Meta AI (FAIR). [7]University of Buffalo, SUNY. [8]University of Chicago and Together Computer.

## Abstract

Recent advances in attention-free sequence models rely on convolutions as alternatives to the attention operator at the core of Transformers. In particular, *long* convolution sequence models have achieved state-of-the-art performance in many domains, but incur a significant cost during auto-regressive inference workloads – naively requiring a full pass (or *caching* of activations) over the input sequence for each generated token – similarly to attention-based models. In this paper, we seek to enable $\mathcal{O}(1)$ compute and memory cost per token in any pre-trained long convolution architecture to reduce memory footprint and increase throughput during generation. Concretely, our methods consist in extracting low-dimensional linear state-space models from each convolution layer, building upon rational interpolation and model-order reduction techniques. We further introduce architectural improvements to convolution-based layers such as Hyena: by weight-tying the filters across channels into *heads*, we achieve higher pre-training quality and reduce the number of filters to be distilled. The resulting model achieves $10\times$ higher throughput than Transformers and $1.5\times$ higher than Hyena at 1.3B parameters, without any loss in quality after distillation.

## 1   Introduction

Attention-free approaches such as *long convolution sequence models* (LCSMs), e.g., H3 [1], Hyena [2], have shown promise in matching Transformer [3, 4] performance across a wide range of tasks, with sub-quadratic complexity with respect to sequence length. Despite the improved efficiency during training on long sequences, unless the convolution filters are either *short* or admit a *low*-dimensional state-state-space realization, LCSMs still need to process the entire growing sequence at every step of auto-regressive generation, similarly to Transformers.

In this work, we seek to refine LCSMs in both **efficiency** and **quality**. First, we study the inference stage, and propose methods to enable a *recurrent* mode for auto-regressive generation. Recurrent modes prescribe the existence of a *state* encoding the past information of the process in a fixed-dimension memory, enabling **constant per-step time** and **constant-memory** in generation. Then, we draw upon an analysis of pre-trained models to develop architectural enhancements for the Hyena block, simultaneously improving model quality and efficiency of the distillation procedure.

**Distilling fast recurrences**   We introduce LaughingHyena, the first distillation approach for LCSMs that enables recurrent inference without impacting downstream quality. LaughingHyena seeks compact recurrences in the form of *state-space models* (SSMs) [5, 6] as the solution of a nonlinear interpolation problem involving the convolution filters of a pre-trained model. Since the total memory cost of SSMs grows linearly in the state dimension $d$, our distillation procedure enables high throughput by enabling processing of large batches during generation. We identify and address three core challenges related to distillation, including the identification of:

- **Target state dimension:** we identify candidate state dimensions of our distilled SSMs by analyzing the spectrum of the Hankel operator associated with each convolution [7].

- **Parametrization:** we address issues with naive parametrizations by introducing a factorized *modal* form, inspired by barycentric [8] and Prony-like [9] methods .

- **Approximation metric:** to ensure compatibility with any downstream task, we choose discrepancy metrics on the convolution filter, rather than model outputs.

In auto-regressive workloads, LaughingHyena-distilled models with state dimension $d$ can generate $K$ tokens in $\mathcal{O}(dK)$ time and with constant $\mathcal{O}(d)$ memory – improving over the $\mathcal{O}(K^2)$ time and $\mathcal{O}(K)$ memory usage of *kv-cached* Transformers and naively executed long convolutions. At model sizes above one billion parameters, LaughingHyena achieves $10\times$ higher peak throughput over comparable Transformers (Figure 1.1), and can process larger batch sizes. Constant memory generation enables larger $K$ for a given a memory constraint e.g., generating 512 tokens with LaughingHyena requires $3\times$ less memory than with a Transformer. At smaller batch sizes, latency of LaughingHyena is also competitive with Transformers, reaching $\geq 2\times$ speedups at longer prompt lengths.

**Improving pre-training quality**  We leverage our analysis of the distillation process to open up new avenues of improvement for LCSM architectures. Indeed, the high compression rates achievable through LaughingHyena hint at sub-utilization of the convolution. We revisit the multi-headed design of H3 [1]; tying weights across channels pushes long convolution filters towards larger effective dimension, and as an additional advantage reduces the runtime of post-training distillation and inference memory footprint. Further, multi-head Hyena models improve on pre-training perplexity over regular Hyena and GPT [10] architectures on the language dataset THE PILE [11].

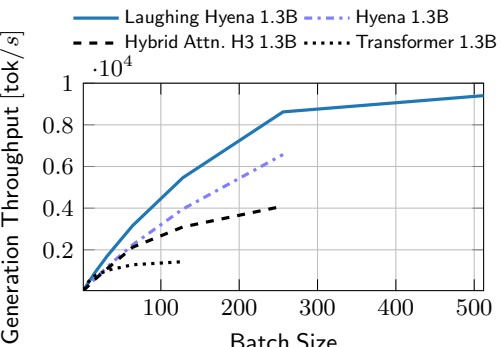

Figure 1.1: Throughput (in generated tokens) of Transformers, H3 and Hyena models. LaughingHyena is a recurrent model distilled from a pre-trained Hyena. Workload involves generating 256 tokens given a prompt of length 512.

## 2  Preliminaries and Related Work

We discuss convolutions, state spaces and auto-regressive generation workloads for sequence models.

**Convolutions**  Let $*$ denote the convolution operator. It is defined as the dual operation to pointwise multiplication under Fourier transform. In signal processing and deep learning alike, one often encounters the causal linear convolution of a filter $h$ (which may extend indefinitely) with an input $u$ of length $L$:

$$(h * u)_t = \sum_{j=0}^{t} h_{t-j} u_j. \tag{2.1}$$

Generally, $u_t \in \mathbb{R}^D$ where $D$ is the width of the signal – or in deep learning parlance – the number of *channels*. Without loss of generality, we specialize our analysis to *single input single output* layers, i.e. with $D = 1$. For the input-output relations of type (2.1), we use the terms *convolution layer* and *linear system* interchangeably. Similarly, the function $t \mapsto h_t$ is referred to as both the *filter* and the *impulse response* of a linear system. Existing convolution sequence models can be classified in terms of the parametrization used for their filters. The class of *implicit* convolutions represent the filter as a parametric function $\gamma_\theta : t \mapsto h_t$.

**State-space realization**  One option is to select $\gamma_\theta$ as the *impulse response* function of a discrete linear time-invariant system,

$$\begin{aligned} x_{t+1} &= \mathsf{A}x_t + \mathsf{B}u_t \\ y_t &= \mathsf{C}x_t + h_0 u_t \end{aligned}, \quad t \mapsto h_t = \begin{cases} h_0 & t = 0 \\ \mathsf{C}\mathsf{A}^{t-1}\mathsf{B} & t > 0 \end{cases} \tag{2.2}$$

with *state* $x_t \in \mathbb{R}^d$, *input* $u_t \in \mathbb{R}$, and *output* $y_t \in \mathbb{R}$. The matrices $\mathsf{A} \in \mathbb{R}^{d \times d}$, $\mathsf{B} \in \mathbb{R}^{d \times 1}$, $\mathsf{C} \in \mathbb{R}^{1 \times d}$, and $h_0 \in \mathbb{R}$ are the learnable parameters of the model while the initial state $x_0$ is usually set to zero such that $u \mapsto y$ is a pure convolution. While linear systems (2.2) are the staple of signal processes and control theory, their use as implicit parametrization of convolution filters in deep neural networks have only recently emerged [12, 6]. Other parametrizations [13, 14, 2] select $\gamma_\theta(t)$ as different flavors

of implicit representation neural networks [15, 16]. The latter are generally more powerful in terms of the class of filters they can represent and flexibility during training, at the cost of losing a fixed state dimension.

## 2.1 Long Convolution Sequence Models

The H-family of convolution sequence models – H3 [1] and Hyena [2] – relies on a combination of long convolutions and data-controlled gating to replace attention with sub-quadratic scaling in sequence length[1]. We use the deep learning convention of naming different projections as *query q*, *key k* and *value v*. Let $\mathsf{M}_q$ and

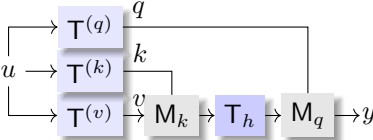

Figure 2.1: H-block. $\mathsf{T}^{(q)}$, $\mathsf{T}^{(k)}$, $\mathsf{T}^{(v)}$ are *short*-convolution operators.

$\mathsf{M}_k$ be the $L$-by-$L$ diagonal matrices whose respective main diagonal entries are the respective entries of length-$L$ sequences $q$ and $k$. A H-block realizes a surrogate attention matrix with a data-controlled, parameterized decomposition in three terms:

$$(q, k, v) \mapsto \mathsf{H}(q,k)v, \quad \mathsf{H}(q,k) = \mathsf{M}_q \mathsf{T}_h \mathsf{M}_k \tag{2.3}$$

where $\mathsf{T}_h \in \mathbb{R}^{L \times L}$ is the Toeplitz matrix constructed from the learnable long convolution filter $h$, i.e., $\mathsf{T}_h = (h_{i-j})_{i,j=0}^{L-1}$. The $qkv$-projections are themselves the output of a convolution between the input sequence and three distinct *short* filters. The degrees of freedom in H-block design are the three short filters[2] and the *long* filter $h$. The long filter can be parameterized using an implicit neural representation [2], state-space model [1], or explicit values [17]. The threefold decomposition of the attention operator, allows evaluation of (2.3) in just $\tilde{\mathcal{O}}(L) := \mathcal{O}(L \log_2 L)$ time (two convolutions[3] and two element-wise products), $y_t = q_t(h * kv)_t$. The overall operator acts on an input $u$ by constructing a third-order multi-variate polynomial of $u$ whose coefficients are controlled (nonlinearly) by parameters of the block.

## 2.2 Auto-Regressive Generation

A typical workload for sequence models is auto-regressive generation. Given a length-$T$ *prompt* $u \in \mathbb{R}^T$, the model is tasked with producing the following $K$ additional outputs – one at a time – for a resulting output sequence $y$ of length $L = T + K$.

**Convolution sequence models** After processing the initial prompt in $\tilde{\mathcal{O}}(T)$ time and obtaining a length-$T$ output $u \mapsto y_0, \ldots, y_{T-1}$, a generic convolution layer can *cache* the output sequence and generate any additional outputs using (2.1) auto-regressively, i.e. $y_{t+1} = \sum_{j=0}^{t} h_{t-j} y_j$ for $t = T-1, \ldots, T+K-1$. It is important to note that auto-regressive generation with generic long convolutions is expensive. It comes with a **quadratic** cost in the number $K$ of tokens to be generated and require storing a cache of length up to $L$.

> **Lemma 2.1.** *Generating $K$ tokens with a long convolution layer* (2.1) *from a length-$T$ prompt has time complexity $\mathcal{O}(T \log_2 T + TK + K^2)$ and requires $\mathcal{O}(L)$ memory.*

**State-space models** When the linear system admits a state space realization (2.2), i.e. it is able to switch between convolution and recurrent mode, the cost of auto-regressive generation can be dramatically reduced. The memory footprint is $\mathcal{O}(d)$: all we need to cache is the state $x_t$, a $d$-dimensional vector. With some further machinery that we develop in next section, we can retain $\tilde{\mathcal{O}}(T)$ time and $\mathcal{O}(T)$ memory to process the prompt[4] and initialize the state $x_{T-1}$. Each additional generation step only requires $\mathcal{O}(d)$ time.

> **Lemma 2.2.** *Generating $K$ tokens with a state-space model* (2.2) *from a length-$T$ prompt has time complexity $\mathcal{O}(T \log_2 T + dK)$ and requires $\mathcal{O}(T + d)$ memory.*

Note that long filters $h$ truncated to length $d$ (i.e. $h_t = 0$ for $t > d - 1$) can also be interpreted as $d$-dimensional SSMs (see Appendix A.7) where the state (a cache) coincides with the last $d$ inputs.

**Transformers** Self-attention is certainly less efficient than long convolutions in processing the prompt, coming with a hefty $\mathcal{O}(T^2)$ time complexity. However, Transformers can achieve a similar

---

[1]In this work, we consider second-order Hyena blocks [2] to automatically extend our findings to H3 [1].

[2]The short filters are *explicitly* parameterized, see [2].

[3]The $qkv$ short convolutions can be evaluated in batch with a single pass. The second convolution is the one with the long filter $h$ and performed via Fast Fourier Transform (FFT), hence the $\tilde{\mathcal{O}}(L)$ complexity.

[4]In §3.4 we show that multiple pre-filling strategies exist, with different trade-offs in time and memory.

efficiency in auto-regressive generation by **caching** the sequences of past keys $\{k_t\}$ and values $\{v_t\}$. Specifically, from $t=T-1$ onward, the new projections $(q_{t+1}, k_{t+1}, v_{t+1})$ are evaluated from the current output $y_t$, and the new output $y_{t+1}$ can be computed in linear time with two reductions

$$y_{t+1} = \frac{\sum_{j=0}^{t+1} \varphi(q_{t+1}k_j)v_j}{\sum_{i=0}^{t+1} \varphi(q_{t+1}k_j)} \quad \text{where } \varphi : \mathbb{R} \to \mathbb{R} \text{ is usually chosen as } \varphi(x) = e^x.$$

**Lemma 2.3.** *Generating $K$ tokens with self-attention from a length-$T$ prompt has time complexity $\mathcal{O}(T^2 + TK + K^2)$ and requires $\mathcal{O}(L)$ memory.*

## 3   The Laughing Hyena Distillery

In this section, we introduce our distillation method. We discuss choosing an approximation objective, a parametrization for the approximant and setting a target state dimension.

Given any pre-trained LCSM, the objective of the distillation procedure is to convert each pre-trained convolution filter into a distinct state-space model (2.2). This should be achieved with the smallest state dimension $d$ which preserves, up to a certain tolerance, the input-output characteristics of the convolution layer. Formally, given a filter $h$ the **distillation problem** is defined as follows.

Given the sequence $h_1, \ldots, h_L$, find a state-space model (2.2) of dimension $d \ll L$, whose input–output behavior *approximates* the one of the convolution with $h$ over the largest class of input sequences.

The choice of approximation metrics and assumptions on the input sequences yield different *distillation objectives*. A *distillation algorithm* constitutes a systematic procedure for optimally choosing the systems matrices with respect to a particular objective. In instances where the original filter $h$ is itself the impulse response of a finite-dimensional state-space model, e.g., when attempting distillation of H3 or S4 [6] filters, the term distillation becomes analogous to *model-order reduction*. Hence, in such cases, the distillation algorithm should yield a state-space representation of a lower order state-dimension.

There exist several algebraic solutions to the model reduction problem [18, 19, 20], typically seeking low-rank structures of the state space by inspecting some invariant of the system, e.g. the *Gramians* in *balanced truncation* [19, Ch. 7]. The lower-order system is then obtained as a projection of the system dynamics onto the found subspace where the system retains desired characteristics, e.g., input-output behavior, stability, etc.

**Truncated filters**   In theory, implicitly parameterized convolution filters can represent arbitrarily long signals. In practice, these filters are trained on a fixed *maximum length $L$*. At inference time the model can then be evaluated for sequences longer than $L$. During distillation it is nonetheless reasonable to treat the pre-trained filters as potentially very long (even beyond $L$) but *finite* impulse response functions [21, 22, 23, 24]. We show how this choice is supported by empirical evidence displaying how pre-trained filters typically decay to zero in finite time (see Appendix D).

***Transfer function* representation**   An alternative description of the system (2.2) is its *transfer function* $H$, defined as the $z$-transform of the impulse response $H(z) = \sum_{t=0}^{\infty} h_t z^{-t}$ for all $z \in \mathbb{C}$ where the sum converges. The transfer function is a *proper rational function* of $z$

$$H(z) = h_0 + \mathsf{C}(z\mathsf{I} - \mathsf{A})^{-1}\mathsf{B} = h_0 + \frac{b_1 z^{-1} + \cdots + b_d z^{-d}}{1 + a_1 z^{-1} + \cdots + a_d z^{-d}}. \tag{3.1}$$

In the $z$-domain, the transfer function defines the input-output map as $Y(z) = H(z)U(z)$. Here, $H(z)$ is defined outside the $\mathbb{C}$-plane circle of radius $\rho(\mathsf{A})$, $\mathbb{D}_{\rho(\mathsf{A})} := \{z \in \mathbb{C} : |z| > \rho(\mathsf{A})\}$ where $\rho(\mathsf{A})$ is the spectral radius of $\mathsf{A}$, i.e. the amplitude of its largest eigenvalue. We can recover all characteristics of a given system equivalently from either its transfer function or state-space representations (see Appendix A.3 for further details and derivations). Notably, the transfer function is an *invariant* of the system: if we apply a change of variables to the state, the transfer function remains unchanged (Lemma A.3). This alone should discourage attempts at modeling filters by learning *dense* state-space matrices $\mathsf{A}, \mathsf{B}, \mathsf{C}$ as such: there are infinitely many equivalent state-space realizations that map to the same system. Starting from coefficients $(a_i)$ and $(b_i)$ of the rational transfer function (3.1), we can compute the impulse response in $\tilde{\mathcal{O}}(L)$ time (Lemma A.6). Moreover, we can map back the transfer function to a special state-space realization – the *companion* canonical form – whose recurrence has time complexity $\mathcal{O}(d)$ (Lemma A.7), compared to

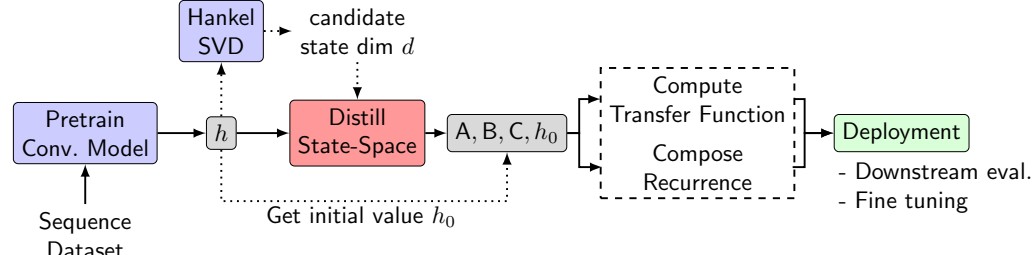

Figure 3.1: The LaughingHyena long convolution sequence model distillation blueprint.

the $\mathcal{O}(d^2)$ of dense state-space matrices. From Lemmas A.3 and A.7 we can also prove that any stable state-space model can be converted by *canonicalization* into its companion form, and thus can be equipped with an efficient recurrence (Thm. A.8).

The distillation problem presents several challenges:

1. **Defining the distillation objective.** A primary decision involves selecting a distillation objective. We are primarily interested in metrics of pure discrepancy between each filter of a pre-trained deep model and its approximator, rather than the expected input-output loss over a distribution of inputs.

2. **Choosing a state-space parametrization.** It is crucial to determine a suitable parametrization of the distilled state-space realization. Once this is decided, the task is to identify the parameters that minimize the distillation desiderata, which can involve challenging optimization problems in itself.

3. **Selecting the target state dimension.** Lastly, a challenge is to estimate the degree to which the model's order can be reduced. In other words, we must select the target state dimension of the distillation process to identify the right trade-off between efficiency and accuracy.

In the following, we address each of these challenges, and provide a comprehensive approach (summarized in Figure 3.1) to distill recurrences from convolution-based architectures.

### 3.1 *Data-Free* Distillation Objectives

We focus on distillation objectives that are independent of the training data and the overall architecture of the neural network under consideration. The distillation loss should be chosen as a pure measure of discrepancy between each convolution filter $h_t$ of the model and their finite-dimensional approximations $\hat{h}_t = \mathsf{CA}^{t-1}\mathsf{B}$. This approach ensures that we do not require a full sequential inference pass over the pre-trained model at each step of distillation procedure and the distilled model can be more broadly applied to downstream tasks. This choice is supported by Young's convolution inequality [25, 26], which indicates that the output approximation error has a bound $\|y - \hat{y}\|_r \leq \|h - \hat{h}\|_q \|u\|_p$ for properly chosen norms[5]. For maximum numerical stability and freedom of parametrization for the approximants, we favor modern unconstrained gradient-based approaches to then solve the resulting distillation program[6]. We design distillation algorithms which either match filters in *time domain* minimizing the $\ell_2$ error ($\|h\|_2 := [\sum_{t \in \mathbb{Z}} |h_t|^2]^{1/2}$) or match their transfer functions optimally with respect to the $\mathcal{H}_2$ norm ($\|H\|_2 := [(1/2\pi) \int_{-\pi}^{\pi} |H(e^{i\omega})|^2 d\omega^{1/2}]$)[7]. As the distillation is carried out via gradient methods, $\ell_2$ is a natural candidate. $\mathcal{H}_2$ error minimization can instead be used to uniformy bound the worst-case discrepancy as $\|h - \hat{h}\|_\infty \leq \|H - \hat{H}\|_2$ (see Appendix A.2 for further details).

### 3.2 **Making** Hyena **Laugh with Modal Interpolation**

Our degrees of freedoms to solve the distillation problem are the matrices A, B, and C of the state-space realization, which determine the filter for all $t > 0$. In distilled SSMs, the passthrough (residual) term cannot be freely assigned: it is simply $h_0$, the value of the original filter at zero. Alternatively, given its

---

[5]$p, q, r > 0$ should satisfy $1/q + 1/p = 1/r + 1$. In the case of infinite sequences defined on the all $\mathbb{Z}$, the norms are taken in a $\ell_p, \ell_q, \ell_r$ sense, respectively. The bound is potentially sharp [27, 28]

[6]For completeness, we also test *balanced* and *modal* truncation techniques on a suite of pre-trained H3 and Hyena models in Appendix E.3.

[7]Such norms are always well-defined for finite sequences of interest which are in $\ell_\infty$.

appealing invariance properties, we can parametrize a proper rational function $\hat{H}(z)$ (3.1) and fit it to the (truncated) transfer function[8] of the original filter $H_L(z) := \sum_{t=0}^{L} h_t z^{-t}$ (see Appendix B.2).

**Modal canonical form** Optimizing the full transfer function can be numerically challenging for several reasons e.g., ensuring stability[9], and ill-posedness for high-order polynomials. A natural solution, inspired by barycentric approaches to rational function approximation [29, 8], is to assume $d$ distinct roots $\lambda_n$ in the denominator's polynomial, $\lambda_n \in \mathsf{roots}(\mathsf{poly}(a))$.

**Proposition 3.1** ([5]). *If* $\mathsf{poly}(a)$ *has distinct roots* $\{\lambda_n \in \mathbb{C}\}$, *then the transfer function of the system can be factorized as* $\hat{H}(z) = \sum_{n=1}^{d} R_n/(z - \lambda_n)$, $\forall z \in \mathbb{D}_{\rho(A)}$ *where* $\{R_n \in \mathbb{C}\}$ *is the residue associated with the pole* $\lambda_n$.

Computing the inverse transform of the expanded transfer function via, e.g., the *Cauchy residue theorem* [30], shows that the resulting impulse response $\hat{h}$ corresponds to a truncated basis of exponentially decaying complex sinusoids

$$\hat{h}_t = \sum_{n=1}^{d} R_n \lambda_n^{t-1}, \quad R_n, \lambda_n \in \mathbb{C}, t > 0. \tag{3.2}$$

In practice, this corresponds to the impulse response of state-space model with diagonal matrix $A = \mathrm{diag}(\lambda_1, \ldots, \lambda_d)$ and such that $B_i C_i = R_i$ for all $i = 1, \ldots, d$. The distillation problem can be then defined in terms of the $L$-point nonlinear least squares interpolation error (squared $\ell_2$) between $h_1, \ldots, h_L$ and (3.2) evaluated for $t = 1, \ldots, L$: $\min_{\{\lambda_n, R_n\}} \|\hat{h} - h\|_2^2$. Note that in case of the target filter $h$ being real-valued, the objective can be replaced by $\|\Re[\hat{h}] - h\|_2^2$.

**Modal Interpolation**

$R_n \lambda_n^{t-1} \bullet h_t$ — $\hat{h}_t$

Figure 3.2: Example of modal interpolation. The approximant is a linear combination of exponentially-decaying complex exponential basis functions with learned decay rate.

Although we find solutions of the distillation (interpolation) problem via modern gradient-based optimization techniques, it is worth mentioning that Prony showed how the nonlinear least square solution can be computed solving two linear problems [9]. However, similar to Padé's method for rational approximation [31], these techniques can be numerically unstable. We opt for a parametrization similar to [32, 33] where each eigenvalue is parameterized in polar form $\lambda_n := A_n e^{i\theta_n}$ and the residues in cartesian form[10]. Note that, with this parametrization we have $\Re[\hat{h}_t] = \sum_n A_n^{t-1}[\Re(R_n)\cos(\theta_n(t-1)) - \Im(R_n)\sin(\theta_n(t-1))]$. We can also solve the distillation problem in the $\mathcal{H}_2$ sense by evaluating $\hat{h}_t$ and $h_t$ at $t = 0, \ldots, L-1$ and taking their respective (discrete) Fourier transform before computing the objective. Efficient evaluation of (3.2) is crucial for distillation. In particular we show the following:

**Lemma 3.1.** *Evaluation of* $(\hat{h}_t)_{t=0}^{L-1}$ (3.2) *can be done in* $\mathcal{O}(dL)$ *time from its modal form and in* $\tilde{\mathcal{O}}(L)$ *time from its proper rational form.*

## 3.3 Minimal Distillation Orders

Distilling into lower-dimensional systems is always desirable as they require fewer parameters to be optimized and they yield recurrences that are (linearly) more efficient in terms of time and memory complexity in post-distillation auto-regressive inference workloads. The dimension of *the smallest possible state-space model with impulse response exactly* $\{h_t\}_{t\in\mathbb{N}}$ is the so-called *McMillan degree* [34]:

$$d^\star = \arg\min_d d \ : \ \exists A \in \mathbb{C}^{d \times d}, B \in \mathbb{C}^{d \times 1}, C \in \mathbb{C}^{1 \times d} \text{ with } h_t = CA^{t-1}B, \forall t > 0 \tag{3.3}$$

**Theorem 3.1** (Ho-Kalman [35, Theorem 2, Corollary]). *Let* $S$ *be the (infinite) Hankel matrix constructed with* $h$, *i.e.* $S := (h_{i+j})_{i,j=1}^{\infty}$. *Then,* $d^\star = \mathrm{rank}(S)$.

---

[8]As already partially discussed in [6], the truncation introduces a correction term in the approximant transfer function. See Appendix A.4.

[9]i.e normalizing denominator polynomial coefficients to constrain roots within the unit circle.

[10]We report additional details on the nuances of the parametrization in Appendix B.1.

A lower bound for $d^\star$ can be estimated from a truncated filter of length $L$ by constructing the $L \times L$ principal sub-matrix $\mathsf{S}_L$ and using the fact that $\mathrm{rank}(\mathsf{S}) \geq \mathrm{rank}(\mathsf{S}_L)$. Inspecting how fast the Hankel singular values $(\sigma_n)_{n=1}^L$ decay in pre-trained convolution models can be predictive of the approximation quality at a fixed dimension. As a rule of thumb, $d$ needs to be sufficiently large for $\sigma_{d+1}$ to be sufficiently small[11]. Specifically, we can prove that the *last* singular value $\sigma_d$ determines the upper bound of distillation quality with a SSM of dimension $d$, in terms of the Hankel norm [19]. This is a direct consequence of Adamjan-Arov-Krein theorem [7] and can be informally stated as follows.

> **Theorem 3.2** (Informal). *Let $h$ be a length-$L$ filter, $\hat{h}$ a distilled filter of order $d < L$ and let $\mathsf{S}_L, \hat{\mathsf{S}}_L$ be the respective Hankel matrices. Then $\inf_{\hat{\mathsf{S}}_L} \|\mathsf{S}_L - \hat{\mathsf{S}}_L\|_2 = \sigma_d$.*

### 3.4 Deploying the Recurrence

Once all the filters of a pre-trained model have been distilled with the proposed modal interpolation technique described above, the model unlocks a *recurrent mode* which allocates a state $x_t \in \mathbb{C}^d$ for each filter and enables fast auto-regressive inference. Deployment of distilled model involves two critical steps: the *pre-filling* and the recurrent update rule itself.

**Fast pre-filling**  During auto-regressive generation, when a length-$T$ prompt is fed to the model, we need to compute the state $x_T$ to start generating new tokens. Using the recurrence, the time complexity of initializing $x_T$ would be $\mathcal{O}(dT)$ with a $\mathcal{O}(d)$ memory footprint. One can alternatively distribute the computation on $d$ processors with a *parallel scan* operation [37, 38] to reach a parallel time complexity $\mathcal{O}(d \log_2 T)$ while incurring in an increased memory requirement of $\mathcal{O}(dT)$[12]. A third option is to use a single FFT convolution to obtain $x_T$ in $\tilde{\mathcal{O}}(T)$ time and $\mathcal{O}(T)$ memory.

> **Proposition 3.2.** $x_T = (\nu_T, \ldots, \nu_{T-d})$ *where $\nu_t = (g * u)_t$ and $g$ is the filter whose transfer function is $1/\mathrm{den}(\hat{H})(z)$ and can be evaluated in $\tilde{\mathcal{O}}(T)$.*

Note that, the fast pre-filling algorithm established by this result requires evaluating the denominator polynomial of $\hat{H}$ from its roots before deployment. This is equivalent to converting the transfer function from its factorized representation to its rational form (3.1).

**Recurrent step**  The update rule is diagonal, thus efficiently evaluated in $\mathcal{O}(d)$ time and memory:

> **Proposition 3.3.** *The filter* (3.2) *has a state space matrices* $\mathsf{A} = \mathrm{diag}(\lambda_1, \ldots, \lambda_d) \in \mathbb{C}^{d \times d}$, $\mathsf{B} = (1, \ldots, 1)^\top \in \mathbb{C}^{d \times 1}$, $\mathsf{C} = (R_1, \ldots, R_d) \in \mathbb{C}^{1 \times d}$ *whose step can be evaluated in $\mathcal{O}(d)$ time and memory.*

As we generally want the output $y_t$ to be real-valued, we can simply update the complex state $x_{t+1} = \mathsf{A}x_t + \mathsf{B}u_t$ and then take the real part of the output, $y_t = \Re[\mathsf{C}x_t] + h_0 u_t$.

## 4  Multi-head Long Convolutions

We can leverage the Hankel spectrum analysis discussed in Section 3.3 to study the dynamics of the effective dimensionality of each convolution filter during LCSMs pre-training. We find that, at initialization, filters correspond to high-dimensional SSMs, and gradually converge to lower-dimensional representations during training. See Appendix E.2 for examples on Hyena and H3 models.

This observation leads to the question: *is it advantageous to perform independent long convolutions on each channel, or can we reduce the total number of filters without loss in quality?* To answer this, we adapt the multi-head layer design proposed by H3 [1] to Hyena [2]:

1. Given the projections $q, k, v \in \mathbb{R}^{L \times D}$, we split them into $M$ chunks of size $N = D/M$, $q^m, k^m, v^m \in \mathbb{R}^{L \times N}$.

2. Each chunk is processed by a modified Hyena operator: first, we perform the outer product of $k^m$ and $v^m$ along the spatial dimension, $z^m := k^m \otimes v^m \in \mathbb{R}^{L \times N \times N}$, apply a long convolution with filter $h^m$ to all $N \times N$ elements independently, then compute $y_t^m = (h^m * z^m)_t q_t^m$, $y^m \in \mathbb{R}^{L \times N}$ as shown in Figure 4.

3. Finally, we compose $y^1, \ldots, y^m$ into a single output $y \in \mathbb{R}^{L \times D}$ via concatenation.

---

[11]Formally, this is related to low-rank approximation characteristics of the Hankel operator; rigorous bounds can be constructed by application of the Eckart–Young–Mirsky theorem [36].

[12]This strategy can also be use to evaluate the filter $\hat{h}$ alternatively to the standard $\mathcal{O}(dL)$ method

An instance of a MultiHyena is equipped with $M < D$ distinct long convolution filters, which leads to $(a)$ faster distillation, with less filters to approximate, $(b)$ lower memory footprint, via a total reduction of the states to cache during generation and $(c)$ faster filter generation, by tying the weights of filter parameters. We note that tying weights of key-value projections has also been shown to be an effective technique to reduce memory cost in Transformers [39, 40].

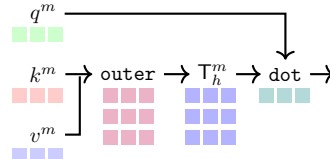

Figure 4.1: A single head of a multi-head Hyena.

Crucially, the multi-head structure of MultiHyena enables us to prove favorable scaling in the *associative recall* synthetic task, which was shown in [2] to be predictive of performance at scale. In associative recall, the model is given a sequence of key-value pairs and a query, and is tasked with matching the query to a key in the sequence by returning its associated value. The difficulty of the task grows with the vocabulary size $s$: larger vocabularies necessitate wider models.

**Theorem 4.1.** *The* MultiHyena *layer, with* $\mathcal{O}(\log s)$ *heads and model size* $\mathcal{O}(\sqrt{s} \log s)$ *can solve the associative recall problem, where $s$ denotes the vocabulary size.*

In Appendix E.1, we empirically verify improved scaling in vocabulary size with multiple heads.

## 5 Experiments

- **Pretraining:** We pretrain a suite of MultiHyena language models on The Pile [11], investigating scaling of perplexity with different amounts of total tokens (5, 10, 15 billion), as well as larger training runs for 300 billion tokens. MultiHyena outperforms Transformers and Hyena.

- **Distillation analysis:** We investigate the relation between optimal distillation orders, Hankel spectrum, and errors on the logits of distilled models.

- **Post-distillation downstreams:** We evaluate the downstream impact of distilling long convolutional language models, reporting HELM [41] and LM-Eval-Harness [42] results.

- **Benchmarking:** We benchmark latency, throughput and memory along the different axes of batch size, sequence length, number of generated tokens. We include base models, distilled models and equivalent Transformers.

### 5.1 Pre-training

To validate the multi-head formulation, we train 150 and 350 million parameter MultiHyena models on The Pile [11] using 8 heads and otherwise the same architecture as equivalent Hyena models, following the setup of [2]. Via the multi-head structure introduced in 4, MultiHyena outperforms both Hyena and Transformers, including on data scaling runs with increasing numbers of tokens and full 300B tokens runs (Table 5.1).

### 5.2 Distillation Analysis

Next, we verify whether Hankel singular values are predictive of downstream errors, and whether large models can be distilled without loss in quality. We apply LaughingHyena distillation to pre-trained MultiHyena, Hyena and H3 of different sizes. Concretely, for each layer and channel of a model, we parametrize the poles $\{\lambda_n\}$ of the modal canonical forms (Section 3.2) at different orders $d$, and solve for each $\ell_2$ approximation problem.

**Approximation errors and spectrum** We investigate the magnitude of approximation errors introduced by LaughingHyena distillation. Given a pretrained MultiHyena model, we compute the errors between original and distilled filters at each layer, averaged across channels. We repeat this process for different distillation orders (state dimension of the model form of Section 3.2). Figure 5.2 visualizes

| Model | PERP. |
|---|---|
| GPT | 9.3 |
| Hyena | 9.3 |
| MultiHyena | **8.7** |

| Model | 5B | 10B | 15B |
|---|---|---|---|
| GPT (125M) | 13.3 | 11.9 | 11.2 |
| Hyena (153M) | 13.3 | 11.8 | 11.1 |
| MultiHyena (153M) | **12.1** | **11.0** | **10.6** |

| Model | 5B | 10B | 15B |
|---|---|---|---|
| GPT (355M) | 11.4 | 9.8 | 9.1 |
| Hyena (355M) | 11.3 | 9.8 | 9.2 |
| MultiHyena (355M) | **10.6** | **9.4** | **8.9** |

Table 5.1: **[Left]** Perplexity of small models on THE PILE, after pre-training for 300 billion tokens. **[Center and Right]** Perplexity on THE PILE for models trained until a total number of tokens e.g., 5 billion (different runs for each token total).

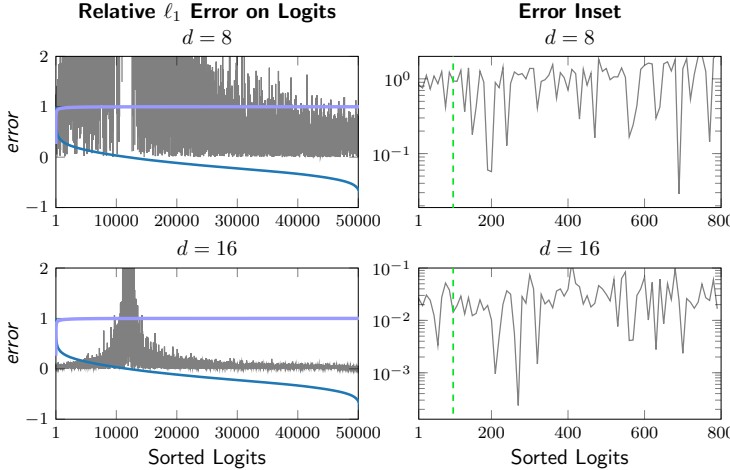

Figure 5.1: Errors between logits of pretrained and distilled MultiHyena. In blue, we plot (ordered) logits, in light blue the cumulative distribution function, and in black the relative errors. The green dotted line indicates the 99.99% percentile. As the errors grow slowly as function of the percentiles, model outputs do not diverge from the base model.

minimum, maximum and average errors, per-layer errors and the distribution of the singular values of the Hankel operator associated to each filter. We observe distillation orders ($> 16$) that yield smalls errors to be predicted by the distribution of singular values. Thus, analysis of the Hankel operator's spectrum is verified to be an effective approach to direct estimation of the optimal distillation order. We also note that the optimal order changes across layers, offering options for further optimization.

**Output errors**  Next, we compute relative $\ell_1$ error between output logits of pre-trained and distilled models to ensure LaughingHyena can be used in generation workloads. The optimal minimal distillation order estimated via Hankel operators (16) is sufficient to keep the output distribution over the vocabulary ($> 50$k entries) close to the pre-trained model, as shown in Figure 5.2. Inspecting the error profile over logits sorted by magnitude reveals our approach to be robust to different sampling strategies for generation, including greedy decoding, top-$k$, top-$p$ [43]. Indeed, the relative errors are $< 10^{-2}$ up to and including the 99.99% percentile of the distribution, meaning e.g., a top-$p$ sampling strategy with large $p$ can be used on a distilled model without drift in outputs (mis-classified tokens). We note that the relative errors are maximum on small-norm logits, which are not required by most sampling strategies. In Appendix D.2, we provide a similar distillation error analysis for Hyena and H3 models. We find that Hyena and can be distilled with less than 32 orders and H3 with less than 8.

### 5.3  Downstream Evaluation

We check how distillation affects downstream performance on language benchmarks. We apply distillation of order 8, 16 and 32 to our THE PILE-pretrained MultiHyena language model and benchmark (Table 5.3) its performance on a suite of canonical (zero shot) tasks from LM-Eval-Harness [42] and HELM [41]. The results are consistent with our error analysis: distillation orders equal or greater to 16 introduce little-to-no quality degradation.

| Model | LAMBADA acc | Winogrande acc | PIQA acc | HellaSwag acc norm. | OpenbookQA acc norm. |
|---|---|---|---|---|---|
| Pythia (160M) | 32.8 | **53.1** | 61.6 | 31.6 | **29.2** |
| MultiHyena (154M) | **43.2** | 52.7 | **64.6** | **34.1** | 29.0 |
| LaughingHyena-16 | 43.1 | 52.6 | 64.7 | 34.1 | 28.9 |
| LaughingHyena-8 | 0.0 | 51.8 | 51.5 | 32.7 | 28.2 |
| LaughingHyena-4 | 0.0 | 49.6 | 53.7 | 26.4 | 26.4 |

Table 5.2: Evaluation of LaughingHyena-distilled models pre and post modal distillation. We test on LM-Eval-Harness [42] and HELM [41] tasks, reporting Pythia [44] performance as a Transformer baseline trained on the same data. LaughingHyena-$d$ is a MultiHyena model with each filter distilled of order $d$.

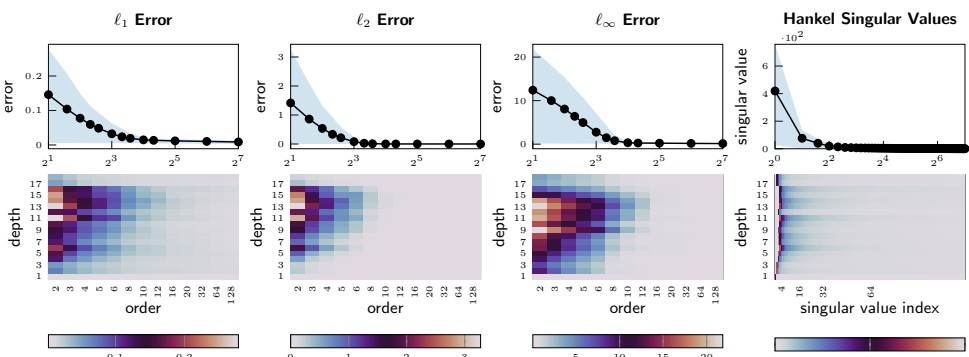

Figure 5.2: Approximation error profiles (min, max, average) on the filters of MultiHyena model after distillation at different orders. We also visualize the distribution of Hankel singular values: if the spectrum decays after $n$ singular values, order $n$ distillation yields low errors.

## 5.4 Benchmarking

We measure throughput, latency and memory usage of LaughingHyena for auto-regressive generation workloads, with initial prompt length $T$ and number of generated tokens $K$. The throughput is computed as number of generated tokens over latency. For each setting (and additional benchmarks), we provide details in Appendix D.4.

**Peak throughput**    Distilled models do not need $kv$-caches. This reduces memory requirement during generation, enabling higher peak throughput in large-batch workloads. We achieve $10\times$ higher throughput than Transformers at size 1.3 billion parameters (Figure 1.1). Throughput is higher than Transformers even at fixed batch sizes, indicating lower latency.

**SSM state dimension and throughput**    For typical distillation orders ($< 100$), peak throughput is not greatly affected. We measure a $2\%$ reduction in throughput from 32 to 64.

**Prompt length**    The throughput of LaughingHyena-distilled models is $4\times$ larger than Transformers at fixed batch size 64 and prompt length 1536 (Figure 5.3). As prompt length increases, the runtime gap between pre-filling via convolutions in LCSMs and pre-filling in Transformers widens (e.g., $\tilde{\mathcal{O}}(T)$ as detailed in Section 3.4, compared to $\mathcal{O}(T^2)$).

**Memory footprint**    Recurrent models do not require $kv$-caches and use constant memory for generation of an arbitrary number of tokens (Figure 5.4).

## 6    Conclusion

We study the efficiency and quality of state-of-the-art long convolutional sequence models. First, we introduce LaughingHyena, a novel distillation method inspired by rational function approximation and model-order reduction techniques. LaughingHyena can be applied after training to extract compact state-space models from each convolutional filter, without loss of quality. Distilled models achieve higher throughput than equivalently-sized Transformers, and can perform auto-regressive generation in constant memory by sidestepping the need to cache previous outputs. We theoretically and empirically investigate the trade-offs of different strategies for fast inference of recurrent models, and introduce architectural improvements to Hyena that improve pretraining quality.

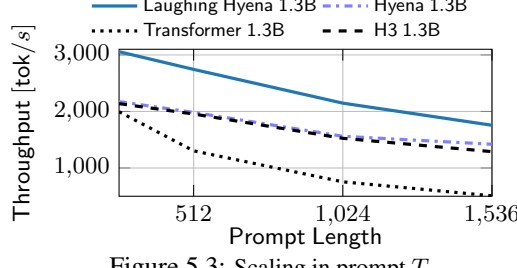

Figure 5.3: Scaling in prompt $T$.

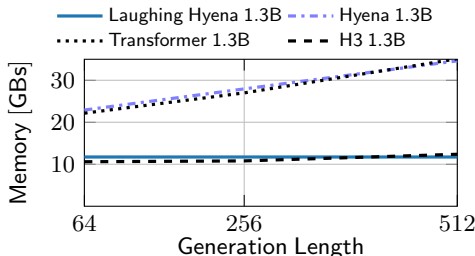

Figure 5.4: Peak GPU memory for generation.

## Acknowledgments

We would like to thank Together Computer for providing the compute used to train models in this paper. We gratefully acknowledge the support of NIH under No. U54EB020405 (Mobilize), NSF under Nos. CCF1763315 (Beyond Sparsity), CCF1563078 (Volume to Velocity), and 1937301 (RTML); US DEVCOM ARL under No. W911NF-21-2-0251 (Interactive Human-AI Teaming); ONR under No. N000141712266 (Unifying Weak Supervision); ONR N00014-20-1-2480: Understanding and Applying Non-Euclidean Geometry in Machine Learning; N000142012275 (NEPTUNE); NXP, Xilinx, LETI-CEA, Intel, IBM, Microsoft, NEC, Toshiba, TSMC, ARM, Hitachi, BASF, Accenture, Ericsson, Qualcomm, Analog Devices, Google Cloud, Salesforce, Total, the HAI-GCP Cloud Credits for Research program, the Stanford Data Science Initiative (SDSI), Department of Defense (DoD) through the National Defense Science and Engineering Graduate Fellowship (NDSEG) Program, and members of the Stanford DAWN project: Facebook, Google, and VMWare. This work is supported by NSF (1651565), AFOSR (FA95501910024), ARO (W911NF-21-1-0125), ONR, DOE (DE-SC0022222), CZ Biohub, and Sloan Fellowship. The U.S. Government is authorized to reproduce and distribute reprints for Governmental purposes notwithstanding any copyright notation thereon. Any opinions, findings, and conclusions or recommendations expressed in this material are those of the authors and do not necessarily reflect the views, policies, or endorsements, either expressed or implied, of NIH, ONR, or the U.S. Government. AR's work is supported by NSF grant# CCF-2247014.

## Broader Impact

In this work, we focus on advances related to efficient models for long sequences.

**Efficiency**    Our distillation methods for constant-memory, high throughput inference in *long convolution sequence models* (LCSMs) can lead to energy savings during model deployment, enabling processing of longer-form content at a fraction of the cost and reducing environmental impact. Improved efficiency may also affect other aspects of AI safety, as it may make it easier produce malicious or harmful content.

**Accessibility**    By improving the efficiency of training and generation,LCSMs and LaughingHyena may contribute to increased accessibility of large language models, lowering the hardware barrier to entry for individuals and organizations with limited resources.

**Steerability**    New method based on LCSMs enable sequence models to process long-form prompts previously inaccessible by Transformers, which may lead to increased control over models via e.g., conditioning on additional instructions [45].

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

# LAUGHING HYENA DISTILLERY
## *Supplementary Material*

## Contents

## Authors Contribution

| | |
|---|---|
| **S.M.** | Conceptualized the research; coordinated collaborations; lead theory development; conducted distillation experiments. |
| **M.P.** | Conceptualized the research; coordinated collaborations; lead the experimental (model pre-training, distillation, benchmarks, downstream evaluation) efforts; coordinated writing and conference submission; optimized inference stack. |
| **D.Y.F.** | Assisted in development of MultiHyena; assisted in pre-training and subsequent benchmarking of distilled models; assisted in writing. |
| **H.K.** | Developed benchmarking suite and interpreted results; assisted in writing. |
| **R.N.P.** | Assisted in theory and algorithmic development; performed model-order reduction experiments of H3 models; assisted in writing. |
| **A.T.** | Conceived and proved Theorem 4.1; assisted in writing. |
| **D.W.R.** | Assisted in Hankel operator spectral analysis; Assisted in writing. |
| **Q.M.** | Assisted in theory development. |
| **B.C.** | Supervised development of benchmarking suite and model deployment. |
| **A.R.** | Supervised theory development (solving associative recall with MultiHyena, Th. 4.1). |
| **C.Z.** | Supervised research; secured compute resources. |
| **C.R.** | Supervised research; reviewed manuscript; secured compute resources. |
| **S.E.** | Supervised research; reviewed manuscript. |
| **Y.B.** | Supervised research; reviewed manuscript. |

*Stefano Massaroli, Michael Poli, and Dan Fu contributed equally to this work. Christopher Ré, Stefano Ermon, and Yoshua Bengio share equal senior authorship.*
All authors read and approved the final manuscript.

## A Linear Systems

### A.1 Extended Notation and System Theory Preliminaries

We first introduce the notation and some mathematical concepts that will be used throughout the paper. By $\mathbb{Z}$ we denote the set of integers, by $\mathbb{R}$ the set of reals, and by $\mathbb{C}$ the set of complex numbers. The variable $t$ stands for *time*. $\ell_p(\mathbb{Z})$ denotes the Banach space of complex-valued sequences $(x_t)_{t \in \mathbb{Z}}$ with finite energy, i.e. $\|x\|_p := [\sum_{t \in \mathbb{Z}} |x_t|^p]^{1/p} < \infty$ for some $1 \leq p < \infty$. $\ell_\infty(\mathbb{Z})$ is instead is the space of sequences for which $\|x\|_\infty := \sup_{t \in \mathbb{Z}} |x_t| < \infty$. With $\mathbb{S}$ denoting the unit circle in the complex plane, $\mathbb{S} := \{z \in \mathbb{C} : |z| = 1\}$ we define $\mathcal{H}_p(\mathbb{S})$ as the space of functions $X$ from $\mathbb{C}$ to itself such that $\|X\|_p := [(1/2\pi) \int_{-\pi}^{\pi} |X(e^{i\omega})|^p d\omega]^{1/p} < \infty$ and $\mathcal{H}_\infty(\mathbb{S})$ the space for which $\|X\|_\infty := \sup_{z \in \mathbb{S}} |X(z)| < \infty$. Particularly, $\mathcal{K}_2(\mathbb{S})$ is a Hilbert space with inner product $\langle X, Y \rangle := (1/2\pi) \int_{-\pi}^{\pi} X(e^{i\omega}) Y^*(e^{i\omega}) d\omega$ where "$*$" denotes complex conjugation. Although we acknowledge we are using the same notation for norms in both $\ell_p(\mathbb{Z})$ and $\mathcal{H}_p(\mathbb{S})$, the correct meaning will always be made clear by the context. The $\mathcal{Z}$-transform of a sequence $x = (x_t)_{t \in \mathbb{Z}}$ is $X(z) = \mathcal{Z}[x](z) := \sum_{t \in \mathbb{Z}} x_t z^{-t}$. We embrace the system theory convention of using capital letters to identify transformed sequences. The $\mathcal{Z}$-transform is a projection of the sequence onto a basis of powers $e_t = r^{-t} e^{i\omega t}$. This basis is not orthogonal unless $r = 1$. That is the basis of the discrete-time Fourier transform $\mathcal{F}$. Hence, $\mathcal{F}$ is defined as $\mathcal{F}[x](e^{i\omega}) = X(e^{i\omega}) := \sum_{t \in \mathbb{Z}} x_t e^{-i\omega t}$. The discrete-time Fourier transform is an isometric isomorphism between $\ell_2(\mathbb{Z})$ and $L_2(\mathbb{S})$. We say that sequences live in the *time domain* and their $\mathcal{Z}$ (or $\mathcal{F}$) transforms in the *frequency domain*.

A *linear system* is a linear operator transforming an input sequence $u$ to an output sequence $y$. If the sequences have continuous support, i.e. $t$ ranges over a continuous set (e.g. $\mathbb{R}$), we have a *continuous-time* system. Conversely, if the sequences have discrete support, i.e. $t$ ranges over a discrete set (e.g. $\mathbb{Z}$), we have a *discrete-time* or *digital* system. **In this manuscript we restrict ourselves to *discrete-time* systems**. Systems can be *single-input single-output* (SISO) if $u$ and $y$ are scalar functions or *multi-input multi-output* if either $u$ or $y$ are vector-valued. **We limit our discussion to SISO systems**. The *impulse response* of a system is the output sequence $y$ when the input sequence $u$ is the Kronecker delta function $\delta_t$ and is usually denoted by the letter $h$. The values $h_t$ of the impulse response sequence are also known as the *Markov parameters* of the system. The most common mathematical representation of a linear system is its convolution form: $y = h * u$, i.e. $y_t = \sum_{j \in \mathbb{Z}} h_{t-j} u_j = \sum_{j \in \mathbb{Z}} h_j u_{t-j}$, $t \in \mathbb{Z}$. In matrix form the input-output relation is given by the Toeplitz operator $\mathsf{T}_h$ corresponding to the (possibly infinitely long) sequence $h$, i.e. $y = \mathsf{T}_h u$. Taking the $\mathcal{Z}$-transforms, we can write the input-output relation as $Y(z) = H(z)U(z)$ (this is just the Fourier convolution theorem extended outside the unit circle). $H(z)$ is called the *transfer function* of the system. When $z = e^{i\omega}$, $H(e^{i\omega})$ is just the discrete-time Fourier transform of $h$ which is called the *frequency response* of the system. A linear system is *causal* if $h_t = 0$ for $t < 0$. A system is called *stable* if the $\mathsf{T}_h$ is a bounded operator. If $u, y \in \ell_2(\mathbb{Z})$, then stability implies $h \in \ell_\infty$. **In the following, we mainly focus on causal stable systems**.

### A.2 Systems Norms

When quantitatively characterizing linear systems, several norms play a crucial role. These norms provide measures of various characteristics of the systems, which are essential in both analysis and filter design.

**The $\ell_2$ and $\mathcal{H}_2$ norms** As defined above, the $\ell_2$ norm represents the *energy* of a signal $h$,
$$\|h\|_2 := \Big[ \sum_{t \in \mathbb{Z}} |h|_t^2 \Big]^{1/2}$$
while $\mathcal{H}_2$ is the energy of the (continuous) spectrum of $h$,
$$\|H\|_2 := \Big[ \frac{1}{2\pi} \int_{-\pi}^{\pi} |X(e^{i\omega})|^2 d\omega \Big]^{1/2}$$
By Parseval's theorem, the $\ell_2$ and $\mathcal{H}_2$ norms are equal, $\|h\|_2 = \|H\|_2$. Further these norms are useful to study the approximation of convolutional filter. The following holds:

> **Lemma A.1** ($\ell_\infty$ output error). *Consider the class of $\ell_2$ measurable inputs such that $\|u\|_2 \leq \zeta$, then for all $H, \hat{H} \in \mathcal{H}_2$,*
> $$\|y - \hat{y}\|_\infty \leq \zeta \|H - \hat{H}\|_2$$

*Proof.*

$$\sup_{t>0} |y_t - \hat{y}_t| = \sup_{t>0} \left| \frac{1}{2\pi} \int_{-\pi}^{\pi} \left[ Y(e^{i\omega}) - \hat{Y}(e^{i\omega}) \right] e^{i\omega t} \mathrm{d}\omega \right|$$

$$\leq \frac{1}{2\pi} \int_{-\pi}^{\pi} |Y(e^{i\omega}) - \hat{Y}(e^{i\omega})| \mathrm{d}\omega$$

$$= \frac{1}{2\pi} \int_{\pi}^{\pi} |H(e^{i\omega}) - \hat{H}(e^{i\omega})||U(e^{i\omega})| \mathrm{d}\omega$$

$$\leq \left[ \frac{1}{2\pi} \int_{-\pi}^{\pi} |H(e^{i\omega}) - \hat{H}(e^{i\omega})|^2 \mathrm{d}\omega \right]^{1/2} \left[ \frac{1}{2\pi} \int_{-\pi}^{\pi} |U(e^{i\omega})|^2 \mathrm{d}\omega \right]^{1/2} \quad \text{Hölder Inequality}$$

$$\leq \left[ \frac{1}{2\pi} \int_{-\pi}^{\pi} |H(e^{i\omega}) - \hat{H}(e^{i\omega})|^2 \mathrm{d}\omega \right]^{1/2} \|u\|_2 \quad \text{Parseval Theorem}$$

$$\leq \zeta \|H - \hat{H}\|_{\mathcal{H}_2}$$

$\square$

If $u$ is the unit impulse function $u_t = \delta_t$ then $\zeta = 1$. The results also holds for finite sequences of length $L$ using the discrete Fourier transform.

> **Lemma A.2** (Impulse response error on finite sequences). *Consider filters $h, \hat{h}$ with finite length $L$. Then, the following holds.*
> $$\|h - \hat{h}\|_\infty \leq \|H - \hat{H}\|_2$$
> *where $H$ and $\hat{H}$ denote the discrete Fourier transforms of $h$ and $\hat{h}$, respectively.*

*Proof.*

$$\|y - \hat{y}\|_\infty := \sup_{t>0} |y_t - \hat{y}_t| = \sup_{t>0} \left| \frac{1}{2\pi} \sum_{n=0}^{L-1} \left[ Y_n - \hat{Y}_n \right] e^{i2\pi nt/L} \right|$$

$$\leq \frac{1}{2\pi} \sum_{n=0}^{L-1} |Y_n - \hat{Y}_n|$$

$$= \frac{1}{2\pi} \sum_{n=0}^{L-1} |H_n - \hat{H}_n||U_n|$$

$$\leq \left[ \frac{1}{2\pi} \sum_{n=0}^{L-1} (H_n - \hat{H}_n)^2 \right]^{1/2} \left[ \frac{1}{2\pi} \sum_{n=0}^{L-1} U_n^2 \right]^{1/2} \quad \text{Hölder Inequality}$$

$$\leq \left[ \frac{1}{2\pi} \sum_{n=0}^{L-1} (H_n - \hat{H}_n)^2 \right]^{1/2} \|u\|_2 \quad \text{Parseval Theorem}$$

$$= \|H - \hat{H}\|_2 \quad \text{using } \|u\|_2 = 1$$

$\square$

### A.3 Transfer Function of State-Space Models

The transfer function (3.1) is derived by taking the $z$-transform of input and state, $U(z) = \mathcal{Z}[u](z), X(z) = \mathcal{Z}[x](z)$. Plugging $U(z)$, $X(z)$ in the state equation (2.2), it holds

$$zX(z) = \mathsf{A}X(z) + \mathsf{B}U(z) \Leftrightarrow X(z) = (z\mathsf{I} - \mathsf{A})^{-1}\mathsf{B}U(z)$$

Substituting in the output equation yields

$$Y(z) = \mathsf{C}(z\mathsf{I} - \mathsf{A})^{-1}\mathsf{B}U(z) + h_0 U(z)$$

The transfer function is then defined as

$$H(z) = \frac{Y(z)}{U(z)} = \mathsf{C}(z\mathsf{I} - \mathsf{A})^{-1}\mathsf{B} + h_0. \tag{A.1}$$

**Alternative derivation** The transfer function can also be derived by direct $z$-transform of the impulse response $h_t$ of the system. This derivation is useful to highlight the region of convergence of the transfer

function.

$$H(z) = h_0 + \sum_{t=1}^{\infty} z^{-t} \mathsf{C} \mathsf{A}^{t-1} \mathsf{B} \qquad\qquad h_0 \text{ is pulled out via } h_0 z^0 = h_0$$

$$= h_0 + \mathsf{C} \left[ \sum_{t=1}^{\infty} z^{-t} \mathsf{A}^{t-1} \right] \mathsf{B} \qquad\qquad \text{multiplication distributes over sum.}$$

(A.2)

$$= h_0 + z^{-1} \mathsf{C} \left[ \sum_{t=1}^{\infty} z^{-(t-1)} \mathsf{A}^{t-1} \right] \mathsf{B} \quad \text{multiply by } z/z$$

$$= h_0 + z^{-1} \mathsf{C} \left[ \sum_{t=0}^{\infty} (z^{-1} \mathsf{A})^t \right] \mathsf{B} \qquad \text{change of index and collect like terms}$$

We look at the convergence of the series $\sum_{t=0}^{\infty} \|z^{-1}\mathsf{A}\|_2^t$. We have

$$\|z^{-1}\mathsf{A}\|_2 \le \|z^{-1}\|_2 \|\mathsf{A}\|_2$$

$$= \|r^{-1} e^{-i\omega}\|_2 \|\mathsf{A}\|_2 \qquad \text{using } z := r e^{i\omega} \in \mathbb{C},\ r, \omega \in \mathbb{R}$$

$$\le r^{-1} \|\mathsf{A}\|_2 = r^{-1} \rho(\mathsf{A})$$

The series converges to $1/(1 - r^{-1}\rho(\mathsf{A}))$ if and only if $r^{-1}\rho(\mathsf{A}) < 1$ i.e. for $r > \rho(\mathsf{A})$. Thus, in the exterior of the disk with radius $\rho(\mathsf{A})$, $\mathbb{D}_{\rho(\mathsf{A})} := \{z \in \mathbb{C} : |z| > \rho(\mathsf{A})\}$, $\sum_{t=0}^{\infty} (z^{-1}\mathsf{A})^t$ converges to $(\mathsf{I} - z^{-1}\mathsf{A})^{-1}$ and

$$z \in \mathbb{D}_{\rho(\mathsf{A})} \;\Rightarrow\; H(z) = h_0 + z^{-1} \mathsf{C}(\mathsf{I} - z^{-1}\mathsf{A})^{-1}\mathsf{B} = h_0 + \mathsf{C}(z\mathsf{I} - A)^{-1}\mathsf{B}$$

The transfer function $H(z) = h_0 + \mathsf{C}(z\mathsf{I} - A)^{-1}\mathsf{B}$ of a stable lumped discrete-time system is defined outside the disc in the complex plane that encloses all the eigenvalues of $\mathsf{A}$.

**Invariance of the transfer function**  $H(z)$ as defined in (A.1) is a *proper*[13] rational function of $z$. In case $h_0 = 0$, $H(z)$ is strictly proper and the denominator is monic:

$$H(z) = \frac{b_1 z^{-1} + \cdots + b_d z^{-d}}{1 + a_1 z^{-1} + \cdots + a_d z^{-d}} \tag{A.3}$$

Specifically, the denominator could be derived from $\mathsf{A}$ with $\det(z\mathsf{I} - A)$, and the numerator is $\det(z\mathsf{I} - A + \mathsf{BC}) + \det(z\mathsf{I} - A)$. We provide a detailed derivation below in Section A.6. While state-space representation involves the analysis and synthesis of model matrices $\mathsf{A}, \mathsf{B}, \mathsf{C}$, the transfer function is entirely characterized by the coefficients $a = (a_n)_{n=1}^{d}$, $b = (b_n)_{n=1}^{d}$ of numerator and denominator polynomials. Notably, the transfer function is an *invariant* of the system: if we apply a change of variables to the state, the transfer function remains unchanged.

> **Lemma A.3.** *Coefficients $a, b$ are **invariant** under any invertible change of variables.*

*Proof.* The proof can be found in [5, pp.95] and follows from the definition of *equivalence transformation*. Consider the state-space matrices of under change of variables $\hat{x} = \mathsf{K}x$,

$$\hat{\mathsf{A}} = \mathsf{K}\mathsf{A}\mathsf{K}^{-1}, \quad \hat{\mathsf{B}} = \mathsf{K}\mathsf{B}, \quad \hat{\mathsf{C}} = \mathsf{C}\mathsf{K}^{-1}, \quad \hat{h}_0 = h_0.$$

The resulting transfer function $H(z)$ can then be computed as

$$\hat{H}(z) = \hat{\mathsf{C}}(z\mathsf{I} - \hat{\mathsf{A}})^{-1}\hat{\mathsf{B}} + \hat{h}_0 = \mathsf{C}\mathsf{K}^{-1}[\mathsf{K}(z\mathsf{I} - A)\mathsf{K}^{-1}]^{-1}\mathsf{K}\mathsf{B} + h_0 = H(z)$$

$\square$

## A.4 Truncated Transfer Functions

In the case of generic *truncated* (finite) impulse response filters, such that $h_t = 0$ for all $t$ greater than a certain value $L$ (which we refer to as the *length* of the filter), the transfer function is simply a polynomial in the complex variable $z$ of order $L$, i.e.

$$H(z) = \sum_{t=0}^{\infty} h_t z^{-t} = \sum_{t=0}^{L} h_t z^{-t} = h_0 + h_1 z^{-1} + \cdots + h_L z^{-L} \tag{A.4}$$

In case the filter is generated by a finite dimensional (lumped parameters) system, i.e. $h_t = \mathsf{C}\mathsf{A}^{t-1}\mathsf{B}$ $t = 1, \ldots, L$, then (A.4) can still be represented exactly by a rational function of order $d$.

---

[13] i.e. such that the denominator's order is not less than the numerator's one.

**Lemma A.4** (Truncated rational transfer functions). *Consider the $L$-truncated impulse response $h_t \in \ell_2(\mathbb{N})$ of a lumped-parameter filter $(\mathsf{A}, \mathsf{B}, \mathsf{C}, h_0)$,*

$$h_t = \begin{cases} h_0 & t = 0 \\ \mathsf{C}\mathsf{A}^{t-1}\mathsf{B} & 1 \le t \le L \\ 0 & t > L \end{cases}.$$

*Then its truncated transfer function is*

$$H_L(z) = \mathcal{Z}\{h\}(z) = h_0 + \mathsf{C}(\mathsf{I} - z^{-L}\mathsf{A}^L)(z\mathsf{I} - \mathsf{A})^{-1}\mathsf{B}$$

*Proof.* By definition of $z$-transform we have

$$H_T(z) = \sum_{t=0}^{\infty} h_t z^{-t} = h_0 + \sum_{t=1}^{L} z^{-t}\mathsf{C}\mathsf{A}^{t-1}\mathsf{B}$$

$$= h_0 + \mathsf{C}\left[\sum_{t=1}^{L} z^{-t}\mathsf{A}^{t-1}\right]\mathsf{B} = h_0 + z^{-1}\mathsf{C}\left[\sum_{t=0}^{L-1}(z^{-1}\mathsf{A})^t\right]\mathsf{B} \quad\quad (\text{A.5})$$

The sum $\sum_{t=0}^{L-1}(z^{-1}\mathsf{A})^t$ is a partial Neumann series and can be manipulated as follows.

$$\sum_{t=0}^{L-1}(z^{-1}\mathsf{A})^t(\mathsf{I} - z^{-1}\mathsf{A}) = \sum_{t=0}^{L-1}(z^{-1}\mathsf{A})^t - \sum_{t=0}^{L-1}(z^{-1}\mathsf{A})^{t+1}$$

$$= \mathsf{I} - (z^{-1}A)^L.$$

Thus,

$$\sum_{t=0}^{L-1}(z^{-1}\mathsf{A})^t = (\mathsf{I} - z^{-L}\mathsf{A}^L)(\mathsf{I} - z^{-1}\mathsf{A})^{-1},$$

which plugged in (A.5) gives $H_L(z) = h_0 + \mathsf{C}(\mathsf{I} - z^{-L}\mathsf{A}^L)(z\mathsf{I} - \mathsf{A})^{-1}\mathsf{B}$, proving the result. □

Because of truncation, evaluating the transfer function $H_L(z)$ on the $L$ *roots of unity* $z = e^{i\omega_k}$, $w_k = 2\pi k/T$ for $k = 0, \ldots L$ gives the length-$L$ discrete Fourier transform (DFT) of the filter:

$$\bar{H}_k := H_L(e^{i\omega_k}) = \sum_{t=0}^{L-1} h_t e^{-i2\pi k/L}, \quad k = 0, \ldots, L-1.$$

In practice, this means that $\bar{H} \in \mathbb{C}^L$ is the FFT of $h$, $\bar{H} = \mathsf{FFT}_L[h]$. If we can find an efficient and stable algorithm to evaluate $\bar{H}$ from the system matrices $(\mathsf{A}, \mathsf{B}, \mathsf{C}, h_0)$, then the FFT-based convolution of truncated filter with an input sequence $u \in \mathbb{R}^L$ can be evaluated in $\tilde{O}(L)$ time.

**Reparametrization** Assume training a LCSM equipped with SSM filters with input/target sequences to be all of length $L$ (smaller sequences can be padded with zeros to the maximum length). Thus, for training purposes, we are only interested in evaluating $\bar{H}$ for the FFT-based convolution.
The truncated transfer function $H_L$ is equal to the original one with a correction term $\mathsf{I} - z^{-L}\mathsf{A}^L$ on the numerator polynomial. As already noted in S4 [6], $z^{-L}$ is conveniently equal to one on the roots of unity, $z^{i\omega_k L} = e^{-i2\pi k} = 1$ for all $k = 0, \ldots, L-1$. Hence, the correction term due to truncation becomes constant: $H_k = \mathsf{C}(\mathsf{I} - \mathsf{A}^L)(\exp(-i2\pi k/L)\mathsf{I} - \mathsf{A})^{-1}\mathsf{B}$; in DFT domain the truncated filter behaves as the infinitely long one with a perturbed $\mathsf{C}$ matrix

$$\bar{\mathsf{C}} = \mathsf{C} - \mathsf{C}\mathsf{A}^L$$

If –as assumed– the SSM is stable $\rho(\mathsf{A}) < 1$, $(i)$ the transfer function is defined on the unit circle, term $\mathsf{C}\mathsf{A}^L$ will go to zero exponentially fast as $L \to \infty$ and $\bar{\mathsf{C}} = \mathsf{C}$ (as expected). As advised in [6], it is desirable to parametrize directly $\bar{\mathsf{C}}$; the expensive computation of the correction term $\mathsf{C}(\mathsf{I} - \mathsf{A}^L)$ is never carried out during training. Instead, the real $\mathsf{C}$ matrix can be retrieved for recurrent inference by inverting the correction term $\mathsf{C} = \bar{\mathsf{C}}(\mathsf{I} - \mathsf{A}^L)^{-1}$, always invertible for stable systems although possibly ill conditioned by eigenvalues too close to the stability margin (the unit circle).

### A.5 From Transfer Function to State-Space

Suppose the coefficients of the numerator and denominator polynomials of a proper transfer function $H$ is given:

$$H(z) = \frac{b_0 + b_1 z^{-1} + \cdots + b_d z^{-d}}{1 + a_1 z^{-1} + \cdots + a_d z^{-d}}. \quad\quad (\text{A.6})$$

A state-space representation of the form (2.2) can be rapidly realized in two steps:

1. **Get delay-free path** From (A.6) we first notice that the *bias* term $h_0$ is $h_0 = b_0$. We thus want to isolate $b_0$ from the rest of the numerator. This can be obtained via long division (see §A.5.1) and results in

$$H(z) = \frac{\beta_1 z^{-1} + \cdots + \beta_N z^{-d}}{1 + a_1 z^{-1} + \cdots + a_d z^{-d}} + b_0, \quad \beta_n = b_n - b_0 a_n \tag{A.7}$$

2. **Get state-space matrices** Given the transfer function $H(z)$ with the isolated pass-through coeffient $b_0$ as in (A.7), we can construct the state-space matrices by *companion* canonical realization:

$$\left[\begin{array}{c|c} \mathsf{A} & \mathsf{B} \\ \hline \mathsf{C} & h_0 \end{array}\right] = \left[\begin{array}{ccccc|c} -a_1 & -a_2 & \cdots & -a_{d-1} & -a_d & 1 \\ 1 & 0 & \cdots & 0 & 0 & 0 \\ 0 & 1 & \cdots & 0 & 0 & 0 \\ \vdots & \vdots & \ddots & \vdots & \vdots & \vdots \\ 0 & 0 & \cdots & 1 & 0 & 0 \\ \hline \beta_1 & \beta_2 & \cdots & \beta_{d-1} & \beta_d & b_0 \end{array}\right] \tag{A.8}$$

Details on the complete *a la* [5] derivation can be found in §A.5.2. A linear system with finite-dimensional state can be equivalently characterized: by its state-space matrices $(\mathsf{A}, \mathsf{B}, \mathsf{C}, h_0)$, by its impulse response function $h$, or by the coefficients $a, b$ (or $\beta$) of the transfer function. A fourth representation is its *linear-constant-coefficients difference equation* form

$$y_t = \sum_{j=0}^{d} b_j u_{t-j} - \sum_{n=1}^{d} a_j y_{t-j},$$

typically used in signal processing literature in the theory of *infinite impulse response* filters (see [46]) and known, in the context of system identification of error-in-variables models, as *auto-regressive moving-average* filters [47, 48].

### A.5.1 Isolating the $h_0$-term from Transfer Function by Long division

If the rational transfer function $H(z)$ accounts for the $h_0$ term, then it is simply proper (order of numerator equals the order of denominator), $h_0$ is necessarily $h_0 = b_0$ (the *delay-free* path). Given the transfer function in this form, we can isolate the $b_0$ term and the strictly rational term of (A.3) by long division. We start by expanding the fraction as

$$H(z) = \frac{q(z)}{p(z)} = \frac{b_0}{p(z)} + \frac{b_1 z^{-1} + \cdots + b_d z^{-d}}{p(z)}.$$

and

$$\frac{b_0}{p(z)} = \frac{b_0 z^d}{z^d + a_1 z^{d-1} + \cdots + a_d}$$

We then use the long division method to compute $b_0/p(z)$:

$$z^d + a_1 z^{d-1} + \cdots + a_d \overline{)\begin{array}{l} b_0 \\ b_0 z^d \\ \underline{b_0 z^d + b_0 a_1 z^{d-1} + \cdots + b_0 a_d} \\ \quad - b_0 a_1 z^{d-1} - \cdots - b_0 a_d \quad \text{(reminder)} \end{array}}$$

to finally get

$$H(z) = b_0 - \frac{b_0 a_1 z^{d-1} + \cdots + b_0 a_d}{z^d + a_1 z^{d-1} + \cdots + a_d} + \frac{b_1 z^{-1} + \cdots + b_d z^{-d}}{p(z)}$$

$$= b_0 + \frac{(b_1 - b_0 a_1)z^{-1} + \cdots + (b_d - b_0 a_d)z^{-d}}{1 + a_1 z^{-1} + \cdots + a_d z^{-d}}$$

Note that the coefficients $b_n$ in (A.3) correspond to $b_n - b_0 a_n$ in (A.6), $b_n \leftarrow b_n - b_0 a_n$. It is indifferent to parameterize the coefficients of the transfer function in either forms. However, if we choose the simply proper representation (A.6), we need to apply the derived correction factor to the numerator coefficients when we separate the $h_0$ term and strictly proper part of $H(z)$.

### A.5.2 Construction of the State-Space from the Transfer Function

**Chen's derivation** The derivation is based on the steps reported for the continuous-time *multi-input multi-output* case in [5]. First, we define a pseudo-state $v$ such that

$$p(z)V(z) = U(z) \quad \Leftrightarrow \quad V(z) = \frac{1}{p(z)}U(z). \tag{A.9}$$

Then, we define the state $x_t := (x_t^1, \ldots, x_t^d) \in \mathbb{R}^d$ as

$$x_t = (v_{t-1}, v_{t-2}, \cdots, v_{t-d}) \quad \Leftrightarrow \quad \mathcal{Z}\{x\}(z) = X(z) = \begin{bmatrix} z^{-1} \\ \vdots \\ z^{-d} \end{bmatrix} V(z). \tag{A.10}$$

From (A.9) we have

$$V(z) + a_1 z^{-1} V(z) + \cdots + a_d z^{-d} V(z) = U(z) \quad \Leftrightarrow$$

$$V(z) = -a_1 z^{-1} V(z) - \cdots - a_d z^{-d} V(z) + U(z) \quad \Leftrightarrow$$

$$v_t = -a_1 v_{t-1} - \cdots - a_d v_{t-d} + u_t \quad \Leftrightarrow \quad \text{time-delay prop. of } \mathcal{Z}\text{-transform}$$

$$x_{t+1}^1 = -a_1 x_t^1 - \cdots - a_d x_t^d + u_t \quad \Leftrightarrow \quad \text{by def. of state (A.10).}$$

Thus, we have the overall recurrence

$$x_{t+1}^1 = -a_1 x_t^1 - \cdots - a_d x_t^d + u_t$$

$$x_{t+1}^2 = x_t^1$$

$$\vdots$$

$$x_{t+1}^d = x_t^{d-1}$$

which can be written in matrix form as

$$x_{t+1} = \begin{bmatrix} -a_1 & -a_2 & \cdots & -a_N \\ 1 & 0 & \cdots & 0 \\ 0 & 1 & \cdots & 0 \\ \vdots & \vdots & \ddots & \vdots \\ 0 & 0 & \cdots & 1 \end{bmatrix} x_t + \begin{bmatrix} 1 \\ 0 \\ \vdots \\ 0 \\ 0 \end{bmatrix} u_t$$

The output spectrum is then given by

$$Y(z) = H(z)U(z) = \frac{q(z)}{p(z)} U(z) + b_0 U(z)$$

$$= q(z)V(z) + b_0 U(z) \qquad \text{by def. of } V(z).$$

Therefore,

$$Y(z) = q(z)V(z) + b_0 U(z) = \begin{bmatrix} \beta_1 & \beta_2 & \cdots & \beta_N \end{bmatrix} \begin{bmatrix} z^{-1} \\ z^{-2} \\ \vdots \\ z^{-d} \end{bmatrix} V(z) + b_0 U(z)$$

$$= \begin{bmatrix} \beta_1 & \beta_2 & \cdots & \beta_d \end{bmatrix} X(z) + b_0 U(z)$$

and the output equation in time-domain is given by

$$y_t = \begin{bmatrix} \beta_1 & \beta_2 & \cdots & \beta_d \end{bmatrix} x_t + b_0 u_t.$$

yielding state-space matrices (A.8).

## A.6 From State-Space to Transfer Function

We detail an implementation oriented method to compute the coefficients $(a_n)_{n=1}^d, (b_n)_{n=0}^d$ of a SSM's transfer function. Recall that

$$H(z) = \mathsf{C}[z\mathsf{I} - \mathsf{A}]^{-1}\mathsf{B} + h_0 = \frac{\mathsf{C}\,\mathrm{Adj}(z\mathsf{I} - \mathsf{A})\mathsf{B} + \det(z\mathsf{I} - \mathsf{A})h_0}{\det(z\mathsf{I} - \mathsf{A})} \tag{A.11}$$

Hence, the denominator coefficients $(a_n)_{n=1}^d$ are simply the coefficients of the characteristic polynomial of matrix A. They can be easily obtained by 1. computing the eigenvalues of A and 2. calculating the coefficients of the polynomial whose roots are such eigenvalues. On the other hand, the numerator apparently involves more complex symbolic manipulation. This can be simplified recalling a classic matrix-determinant identity:

**Lemma A.5** ([49]). *Let* M, B, *and* C *respectively denote matrices of orders* $d \times d$, $d \times 1$, *and* $1 \times d$. *Then,*

$$\det(\mathsf{M} + \mathsf{BC}) = \det(\mathsf{M}) + \mathsf{C}\,\mathrm{Adj}(\mathsf{M})\mathsf{B}.$$

Applying Lemma A.5 to (A.11) we obtain

$$H(z) = \frac{\det(z\mathsf{I} - \mathsf{A} + \mathsf{BC}) + \det(z\mathsf{I} - \mathsf{A})(h_0 - 1)}{\det(z\mathsf{I} - \mathsf{A})}.$$

Let poly$(r)$ denote the coefficients of the polynomials with roots $r = (r_1, \ldots, r_d)$. Then $a = \text{poly}(\text{eig}(A))$. Since A and $A - BC$ are of equal dimension, their characteristic polynomials have equal order and therefore

$$b = \text{poly}(\text{eig}(A - BC)) + \text{poly}(\text{eig}(A))(h_0 - 1)$$

Listing 1: State-space $\to$ transfer function conversion code

```
def get_tf_from_ss(A,B,C,h0):
    a = poly(eig(A))
    b = poly(eig(A — outer(B,C))) + (h0—1)*a
    return a, b
```

### A.7  State-Space Representation of Truncated Filters.

A truncated filter $h_0, \ldots, h_L$ – as the ones found in any standard convolutional neural network – can be represented by a $L$-dimensional companion canonical SSM. The filter's transfer function $H(z) = h_0 + h_1 z^{-1} + \cdots + h_L z^{-L}$ is polynomial, i.e. a rational function with the denominator's coefficients set to zero. Following the canonical realization process detailed in Section A.5, the truncated filter has state-space form:

$$x_{t+1} = \begin{bmatrix} 0 & 0 & \cdots & 0 & 0 \\ 1 & 0 & \cdots & 0 & 0 \\ 0 & 1 & \cdots & 0 & 0 \\ \vdots & \vdots & \ddots & \vdots & \vdots \\ 0 & 0 & \cdots & 1 & 0 \end{bmatrix} x_t + \begin{bmatrix} 1 \\ 0 \\ \vdots \\ 0 \\ 0 \end{bmatrix} u_t$$

$$y_t = \begin{bmatrix} h_1 & h_2 & \cdots & h_L \end{bmatrix} x_t + h_0 u_t.$$

If $x_0 = \mathbb{0}_L$ and $u_t = 0$ for negative $t$, then at each $t > 0$ the state is a shifted copy of the input sequence $x_t = (u_{t-1}, \ldots, u_{t-L}) \in \mathbb{R}^L$. Nonetheless, the asymptotic complexity of computing one recurrent step is $\mathcal{O}(L)$ as it requires only a shift operation and a length-$L$ dot product

$$\begin{aligned} x_{t+1}^1 &= u_t \\ x_{t+1}^{2:L} &= \text{shift}(x_t) \\ y_t &= \langle h_{1:L}, x_t \rangle + h_0 u_t. \end{aligned} \tag{A.12}$$

The memory footprint is also $\mathcal{O}(L)$. In [1] it is proposed the use of shift-type SSMs to parametrize one of the filters of the H3 block.

### A.8  Efficient Computation of State-Space Models

#### A.8.1  Fast Evaluation of the Transfer Function

Computing $H(z)$ at any point $z \in \mathbb{C}$ concerns the evaluation of the $d$-order polynomial of numerator and denominator,

$$H(z) = \frac{q(z)}{p(z)} = \frac{\sum_{n=1}^{d} b_n z^{-n}}{1 + \sum_{n=1}^{d} a_n z^{-n}}$$

In practice, we are mainly interested in a fast algorithm that allows computing $H$ on the $L$ roots of unity to obtain the DFT of the filter. The DFT of the filter can be then readily used to perform a FFT-based convolution with a length-$L$ input sequence $u$ or to recover the impulse response function via inverse DFT. We prove the following:

> **Lemma A.6.** *Given the coefficients $a$, $b$ of the transfer function, the frequency and impulse response of the filter can be evaluated in $\tilde{\mathcal{O}}(L)$ time.*

*Proof.* The result is proven showing that the transfer function can be evaluated in $\tilde{\mathcal{O}}(L)$ time on the $L$ roots of unity. The fastest method to evaluate polynomials on $L$ arbitrary points $z$ of the complex plane is generally the Horner's scheme. This method is based on a sequence of nested multiplications and computes the polynomial from its vector of coefficients, delivering a time complexity of $\mathcal{O}(dL)$. More explicitly, Horner's scheme determines $p(z)$ as $p(z) = ((\cdots((a_d z^{-1} + a_{d-1}) z^{-1} + a_{d-2}) \cdots) z^{-1} + a_2) z^{-1} + a_1) z^{-1} + 1$. Each step involves a multiplication and an addition, making a total of $2d$ operations per evaluation point. Thus, for $L$ points, the total number of operations amounts to $\mathcal{O}(dL)$.
Effectively, Horner's approach implements the matrix-vector product of an $L$-by-$(d+1)$ Vandermonde matrix $V \in \mathbb{C}^{L \times (d+1)}$ constructed by $L$ evaluation points $(z_0, \ldots, z_{L-1})$ with the vector of coefficients

$a = (1, a_1, \ldots, a_d)^\top$:

$$\begin{bmatrix} p(z_0) \\ p(z_1) \\ \vdots \\ p(z_{L-1}) \end{bmatrix} = \begin{bmatrix} 1 & z_0^{-1} & z_0^{-2} & \cdots & z_0^{-d} \\ 1 & z_1^{-1} & z_1^{-2} & \cdots & z_1^{-d} \\ \vdots & \vdots & \vdots & \ddots & \vdots \\ 1 & z_{L-1}^{-1} & z_{L-1}^{-2} & \cdots & z_{L-1}^{-d} \end{bmatrix} \begin{bmatrix} 1 \\ a_1 \\ \vdots \\ a_d \end{bmatrix} = \mathsf{V} a$$

Significantly, if the polynomial is required to be evaluated at the roots of unity, the Vandermonde matrix simplifies corresponds to the $L \times (d+1)$ DFT matrix. Further, zero-padding the coefficient vector to length $L$, enables the use a single length-$L$ FFT to compute the matrix-vector product in $\tilde{\mathcal{O}}(L)$ time. Thus, the numerator and denominator polynomials of the transfer function can be evaluated, on the roots of unity, in $\tilde{\mathcal{O}}(L)$ time by taking the FFT of the padded numerator / denominator coefficients $a, b$ and subsequently dividing element-wise the two sequences as $\mathsf{FFT}_L[b]/\mathsf{FFT}_L[a]$. The overall time complexity to obtain the impulse response is also $\tilde{\mathcal{O}}(L)$ since $h$ can be recovered taking an inverse FFT of the frequency response. $\qquad \square$

### A.8.2 Fast Companion Recurrence

The recurrent step of a generic SSM (2.2) with dense system matrices usually requires $\mathcal{O}(d^2)$ operations due to the matrix-vector product $\mathsf{A}x_t$. We show how the recurrence of SSMs in *companion canonical form*, i.e. with system's matrices (A.8), requires only $\mathcal{O}(d)$ operations.

> **Lemma A.7.** *The recurrent step of a state-space model in companion canonical form* (A.8) *can be evaluated in $\mathcal{O}(d)$ time and memory.*

*Proof.* The companion state matrix $\mathsf{A}$ can be broken down into a lower shift matrix $\mathsf{L}_N$ and a low-rank term. Particularly, with $e_1$ the first element of the canonical basis of $\mathbb{R}^N$ and $\alpha = (a_1, \ldots, a_N)$, we have

$$\mathsf{A} = \mathsf{L}_N - e_1 \otimes \alpha.$$

It follows that the recurrent update can be simplified to

$$x_{t+1} = (\mathsf{L}_N - e_1 \otimes \alpha) \, x_t + \mathsf{B}u_t$$
$$y_t = \mathsf{C}x_t + b_0 u_t$$

The peculiarity of this formulation is that we never need to construct the matrices to perform the recurrence. In particular we have:

$$x_{t+1}^1 = u_t - \alpha^\top x_t$$
$$x_{t+1}^{2:N} = \mathsf{shift}(x_t)$$
$$y_t = \beta^\top x_t + b_0 u_t$$

Thus, each step only requires two inner products ($d$ multiplications and $d$ sums each) and one shift operation, totaling $\mathcal{O}(d)$ operations. $\qquad \square$

The proof of Lemma A.7 yields the practical implementation of the recurrence:

Listing 2: Python implementation of the companion canonical recurrence

```python
def step(x, u, alpha, beta, b0):
        y = dot(beta, x) + b0 * u
        lr = u - dot(alpha, x)
        x = roll(x)
        x[0] = lr
        return x, y
```

### A.8.3 Canonization of State-Space Models

The companion canonical form discussed in Section A.5 is the ideal representation to deploy SSM-based convolutional layers: $i)$ it comes with a $\mathcal{O}(d)$ fast recurrence and $ii)$ allows to swiftly switch between time and frequency domains with a direct mapping between state-space matrices and coefficient of the transfer function (which in turn allow $\tilde{\mathcal{O}}(L)$ fast convolutions).

Aside from [50], which directly parametrizes S4 layers in companion canonical form, all the other parameterizations [12, 6, 32, 17, 33] can be *converted* (*canonized*), under mild assumptions.

**Lemma A.8** (Canonization of SSMs). *Any state-space model* (2.2) *with proper transfer function can be converted in companion canonical form.*

*Proof.* The result can be proved following the two-step conversion process.

1. **Get the coefficients of the transfer function**: Given the original state-space matrices $(A, B, C, h_0)$, the transfer function is given by $H(z) = C(zI - A)^{-1}B + h_0$. A proper rational function has the form $H(z) = q(z)/p(z)$ where the numerator $q(z)$ has coefficients $b = (b_n)_{n=0}^d$ and the denominator has coefficients $a = (a_n)_{n=0}^d$ ($a_0 = 1$ since $p$ is monic). As shown in Section A.6, the coefficients of the transfer function can be extracted in closed-form as $b = \mathsf{poly}(\mathsf{eig}(A - BC)) + \mathsf{poly}(\mathsf{eig}(A))(1 - h_0)$ and $a = \mathsf{poly}(\mathsf{eig}(A))$ [14];

2. **Construct companion matrices** Given the coefficients $a$ and $b$ a new set of canonical state-space matrices which realize the transfer function can be obtained following the recipe of Section A.5.

The resulting companion SSM is equivalent the the original one since they share the same transfer function. □

---

[14] $\mathsf{eig}(A)$ contains the eigenvalues of $A$. $\mathsf{poly}(r)$ yields the coefficients of the polynomial whose roots are the elements of $r \in \mathbb{C}^d$.

## B  LaughingHyena: **Further Details**

### B.1  Parametrization of Modal Interpolators

**Complex-conjugate states** Assuming even distilling dimension $d$, we pick poles $\lambda_n$ and residues $\mathbb{R}_n$ in complex-conjugate pairs:

$$\mathsf{A} = \mathrm{diag}(\lambda_1, \cdots, \lambda_{d/2}, \lambda_1^*, \cdots, \lambda_{d/2}^*)$$
$$\mathsf{C} = \frac{1}{2}[R_1, \cdots, R_{d/2}, R_1^*, \cdots, R_{d/2}^*] \tag{B.1}$$

which allow partitioning the state-space matrices as

$$\mathsf{A} = \begin{bmatrix} \lambda & \\ & \lambda^* \end{bmatrix}, \quad \mathsf{C} = \frac{1}{2}\begin{bmatrix} R & R^* \end{bmatrix}, \tag{B.2}$$

where

$$\lambda = \mathrm{diag}(\lambda_1, \cdots, \lambda_{d/2}) \text{ and } R = [R_1, \cdots, R_{d/2}]. \tag{B.3}$$

If we also partition the state as $x = (\bar{x}, \tilde{x})$, $\bar{x}, \tilde{x} \in \mathbb{C}^{d/2}$, the resulting recurrence has the form

$$\bar{x}_{t+1} = \lambda \bar{x}_t + \mathbb{1}_{d/2}\, u_t$$
$$\tilde{x}_{t+1} = \lambda^* \tilde{x}_t + \mathbb{1}_{d/2}\, u_t \tag{B.4}$$

We have

$$\bar{x}_t = \lambda^t \bar{x}_0 + \sum_{j=0}^{t-1} \lambda^{t-j-1} \mathbb{1}_{d/2} u_t, \quad \tilde{x}_t = [\lambda^*]^t \tilde{x}_0 + \sum_{j=0}^{t-1} [\lambda^*]^{t-j-1} \mathbb{1}_{d/2} u_t \tag{B.5}$$

Thus, if $\tilde{x}_0 = \bar{x}_0^*$, then $\tilde{x}_t = \bar{x}_t^*$ for all $t > 0$. Hence, at inference time we only need to propagate forward half of the state – say $\bar{x}$ – and then compute the output as

$$y_t = \mathsf{D}u_t + \frac{1}{2}(R\bar{x}_t + R^*\bar{x}_t^*)$$
$$= \mathsf{D}u_t + \mathfrak{R}\{R\bar{x}_t\} \tag{B.6}$$
$$= \mathsf{D}u_t + \mathfrak{R}\{R\}\mathfrak{R}\{\bar{x}_t\} - \mathfrak{I}\{R\}\mathfrak{I}\{\bar{x}_t\}$$

This parametrization allows to update only half of the state, reducing the time and memory cost compared to an *unconstrained* linear system with complex coefficients. However, the *implicitly* achieved realness of the output (assuming $\mathsf{D} = h_0 \in \mathbb{R}$ and $u_t \in \mathbb{R}$) comes at a cost of expressivity: such a system is equivalent to an unconstrained complex linear system of dimension $d/2$ of which we only keep the real part of the output.

**Poles and residues** For the modal interpolation, the parametrization is analogous to the one of a diagonal state space model [33]. Poles $\lambda_n$ and residues $R_n$ need both to be complex numbers. In [33] the authors suggest parametrizing real and imaginary components of $\mathsf{B}$ and $\mathsf{C}$ matrices while representing the eigenvalues $\lambda_n$ in polar form, $\lambda_n = r_n e^{i\alpha_n}$ with $r_n$ and $\alpha_n$ being themselves exponential functions of the actual trainable parameters, $r_n = e^{-e^{\nu_n}}$, $\alpha_n = e^{\zeta_n}$ leading to

$$\lambda_n = \exp\{-\exp\{\nu_n\} + i\exp\{\zeta_n\}\}, \quad \nu_n, \zeta_n \in \mathbb{R} \tag{B.7}$$

This ensures stability of the poles $|\lambda_n| < 1$ and positive-only phases $\alpha_n$. For the purpose of distillation we propose a simplified parametrization as follows:

1. We only parametrize the $\mathsf{C}$ vector. Parametrizing both $\mathsf{B}$ and $\mathsf{C}$ is redundant and increases the computational cost of performing each step of the recurrence. The residues $R_n$ correspond in fact to $R_n = \mathsf{C}_n \mathsf{B}_n$ of a diagonal state space model. Setting $\mathsf{B} = \mathbb{1}_d$ saves parameters without harming expressivity. Further if $\mathsf{B}$ is different from $\mathbb{1}_k$ it needs to be multiplied to $u_t$ at each recurrence step. $\mathsf{C}_n = R_n = \mathfrak{R}[R_n] + i\mathfrak{I}[R_n]$ and $\mathfrak{R}[R_n], \mathfrak{I}[R_n]$ are the trainable parameters of the residue.

2. For the purpose of distillation we have no benefit in forcing the eigenvalues of the model to be stable, i.e. constrained to lie strictly inside the unit circle. Instead, such constrain may actually harm the expressivity of the approximant. We choose the the simpler parametrization $\lambda_n = r_n e^{i\alpha_n}$, $r_n, \alpha_n \in \mathbb{R}$.

### B.2  Distillation as Rational Interpolation

**Distillation as *rational interpolation*** Approximating a filter with an SSM can be thus achieved by fitting a proper rational function to the (truncated) transfer function of the original filter $H_L(z) := \sum_{t=0}^{L} h_t z^{-t}$. That is,

$$\text{Find } a, \ b \text{ such that } h_0 + h_1 z^{-1} + \cdots + h_L z^{-L} \approx h_0 + Q_b(z)/P_a(z). \qquad \text{(B.8)}$$

A modern[15] way to solve this problem by $\mathcal{H}_2$ error minimization via gradient descent[16]. We can use the Fast Fourier Transform (FFT) to evaluate both the target and distilled transfer functions and solve:

$$\min_{a,b\in\mathbb{R}^d} \sum_{k=0}^{L} |\mathsf{FFT}_L[h]_k - h_0 - \mathsf{FFT}_L[b]_k/\mathsf{FFT}_L[a]_k|^2. \qquad \text{(B.9)}$$

To ensure stability of the distilled filters and well-conditioned gradient descent dynamics, the roots of the denominator polynomial must strictly lie inside the unit circle ($\rho(\mathsf{A}) < 1$). This, in turn, requires constraining the coefficients $a$ into the region $\{a : \text{poly}(a) \text{ is stable}\}$ which is by itself an open research problem [52, 53]. Experimentally, we observe that standard coefficient normalization techniques overly restrict the parameters space and lead to poor distillation performances at reasonable order.

---

[15]In the late 19th century, Henri Padé had already proposed a closed-form solution of the above problem that achieves $o(z^{-L})$ error for $z \to \infty$ using $L=2d$ samples of the impulse response. His method [31] solves a $L$-dimensional linear problem that, however, is known to often become numerically ill-conditioned even with small $d$ [51]

[16]In the case of finite sequences, the $\mathcal{H}_2$ norm becomes the standard Euclidean metric evaluated on the $L+1$ roots of unity,i.e. $(\sum_{k=0}^{L} |H(e^{i2\pi k/(L+1)})|^2)^{1/2}$.

## C  Proofs

### C.1  Proof of Lemma 2.1

> Generating $K$ tokens with a long convolution layer (2.1) from a length-$T$ prompt has time complexity $\mathcal{O}(T \log_2 T + TK + K^2)$ and requires $\mathcal{O}(L)$ memory.

*Proof.* We compute the time complexity memory of a length-$T$ prompt processing (*pre-filling*) and subsequent auto-regressive decoding of $K$ tokens. The auto-regressive generation of long convolution computes the next token as by

$$t = T, \ldots, T + K - 1 \;\Rightarrow\; y_t = \sum_{j=0}^{t-1} h_{t-j} y_j \tag{C.1}$$

The pre-filling step is needed to prime this recurrence by computing the first $T$ outputs till $y_{T-1}$ from the length-$T$ prompt $u$. This is just a convolution between two length-$T$ signal and requires $\mathcal{O}(T \log_2 T)$ time and linear memory. The auto-regressive decoding of $K$ tokens requires $K$ steps (C.1) with the length of the sequences increasing by 1 at each step. Thus we have a total asymptotic complexity of

$$\sum_{k=0}^{K-1} (T + k) = TK + \frac{1}{2} K(K + 1). \tag{C.2}$$

and requires at worst ($k = K - 1$) to store the length $T + K = L$ generated output sequence, i.e. $\mathcal{O}(L)$ memory. In the limit we thus have a total time complexity of $\mathcal{O}(T \log_2 T + TK + K^2)$ and $\mathcal{O}(L)$ memory. $\qquad\square$

### C.2  Proof of Lemma 2.2

> Generating $K$ tokens with a SSM (2.2) from a length-$T$ prompt has time complexity $\mathcal{O}(T \log_2 T + dK)$ and requires $\mathcal{O}(d)$ memory.

*Proof.* In autoregressive mode, the cost of generating one token is the cost of evaluating the state recurrence (2.2). Each step then requires $\mathcal{O}(d)$ time and memory for the class of SSMs considered in this work (see Lemma A.8). Hence, generating $K$ tokens costs $\mathcal{O}(dK)$ time and constant $\mathcal{O}(d)$ memory (we only need to store the current state).

The recurrence is initialized for autoregressive generation with the post-prompt state $x_{T-1}$ and output $y_{T-1}$. The latter can be recovered in linear time and memory $\mathcal{O}(T)$ by definition $y_{T-1} = \sum_{j=0}^{T-1} h_{t-j} u_j$ (assuming to have the impulse response $h$ available) and state $x_{T-1}$ in $\mathcal{O}(dT)$ time and $d$ memory through the recurrence. The overall asymptotic cost is therefore $\mathcal{O}(dL)$ time and $\mathcal{O}(d)$ memory. $\qquad\square$

Note that, for prompts and SSMs of practical sizes we usually have $d > \log_2 T$. In such a case the state $x_{T-1}$ can be computed in $T \log_2 T$ time rather than $dT$ by Proposition 3.2.

### C.3  Proof of Lemma 2.3

> Generating $K$ tokens with self-attention from a length-$T$ prompt has time complexity $\mathcal{O}(T^2 + TK + K^2)$ and requires $\mathcal{O}(L)$ memory.

*Proof.* The proof is identical to the one of Lemma 2.1, with the only difference of a quadratic asymptotic cost $\mathcal{O}(T^2)$ to process the prompt obtain the $kv$ cache. $\qquad\square$

Self-attention suffers with long contexts: it is significantly more expensive in prefilling than long convolutions and SSMs due to its quadratic cost. Nonetheless, in autoregressive mode, self-attention reaches the same overall asymptotic complexity $\mathcal{O}(TK + K^2)$ as long convolutions (with the memory overhead of having to cache $k$ and $v$).

### C.4  Proof of Proposition 3.1

> If A has semi-simple eigenvalues $\lambda_n \in \mathbb{C}$, then the transfer function of the system can be decomposed as $\hat{H}(z) = \sum_{n=1}^{d} R_n / (z - \lambda_n)$ where $R_n \in \mathbb{C}$ is the residue associated with the pole $\lambda_n$.

*Proof.* If $A$ is semi-simple, then it is diagonalized by a basis $V$ of eigenvectors; it admits an eigenvalue decomposition $\mathsf{diag}(\lambda) = VAV^{-1}$ where $\lambda = (\lambda_1, \ldots, \lambda_d) \in \mathbb{C}^d$ contains the eigenvalues of $A$. Projecting the state onto the basis of eigenvectors, $s := Vx$, the state space model is is transformed into modal form:

$$s_{t+1} = VAV^{-1}s_t + VBu_t \qquad s_{t+1} = \mathsf{diag}(\lambda)s_t + \tilde{B}u_t$$
$$y_t = CV^{-1}s_t \qquad \Leftrightarrow \qquad y_t = \tilde{C}s_t$$

where $\tilde{B} := VB = (\tilde{b}_n)_{n=1}^d$ and $\tilde{C} := CV^{-1} = (\tilde{c}_n)_{n=1}^d$. In modal form, the state equations are decoupled, i.e.

$$s_{t+1}^n = \lambda_n s_t^n + \tilde{b}_n u_t$$
$$y_t = \sum_{n=1}^d \tilde{c}_n s_t^n.$$

Taking the $z$-transform of the output equation and each state equation yields

$$S_n(z) = \mathcal{Z}[s^n](z) = \frac{\tilde{b}_n}{z - \lambda_n}U(z) \ \ n = 1, \ldots, d$$

$$Y(z) = \sum_{n=1}^d \tilde{c}_n S_n(z)$$

Thus, the overall transfer function is

$$H(z) = \frac{Y(z)}{U(z)} = \sum_{n=1}^d \frac{\tilde{b}_n \tilde{c}_n}{z - \lambda_n}$$

Letting $R_n = \tilde{b}_n \tilde{c}_n$, proves the result. $\qquad \square$

## C.5 Proof of Lemma 3.1

> The distilled filter $\hat{h}$ in modal form (3.2) can be computed in $\mathcal{O}(dL)$ time from its modal form and in $\tilde{\mathcal{O}}(L)$ from its rational form.

Recalling (3.2), $\hat{h}_t = \sum_{n=1}^d R_n \lambda_n^{t-1}$, $R_n, \lambda_n \in \mathbb{C}, t > 0$, the $\mathcal{O}(dL)$ complexity of the impulse response is apparent: for each of the $t = 1, \ldots, L$, $\hat{h}_t$ can be computed in $\mathcal{O}(d)$ time.
The $\tilde{\mathcal{O}}(L)$ cost from the rational form follows by Lemma A.6.

## C.6 Proof of Theorem 3.2

> Let $h$ be a length-$L$ filter, $\hat{h}$ a distilled filter of order $d < L$ and let $S_L, \hat{S}_L$ be the respective Hankel matrices. Then $\inf_{\hat{S}_L} \|S_L - \hat{S}_L\|_2 = \sigma_d$.

*Proof.* The theorem characterizes the best-case scenario in terms of approximation error of the distilled SSMs or a certain order $d$ where it is clear that $\mathsf{rank}\,\hat{S} \leq d$. This theorem is a direct application of the Adamyan-Arov-Krein (AAK) theory of infinite Hankel operators [7]. Let $\hat{S}_L^* = \arg\inf_{\hat{S}_L} \|S_L - \hat{S}_L\|_2$; the AAK theorem says that every causal system can be optimally approximated by another causal system of lower dimension. Optimal here means

$$\inf \|S_L - \hat{S}_L^*\| = \inf \|S_L - K\|$$

where the first infimum is taken over all Hankel matrices $\hat{S}_L^*$ and the second over all arbitrary matrices $K$ (see [19, Chapter 8] and [54] for further details and references).
$\qquad \square$

## C.7 Proof of Proposition 3.3

> The filter (3.2) has a state space matrices $A = \mathsf{diag}(\lambda_1, \ldots, \lambda_d) \in \mathbb{C}^{d \times d}$, $B = (1, \ldots, 1)^\top \in \mathbb{C}^{d \times 1}$, $C = (R_1, \ldots, R_d) \in \mathbb{C}^{1 \times d}$, $D = h_0$ whose step can be evaluated in $\mathcal{O}(d)$ time and memory.

D is set to $h_0$ by default. The result is proven showing that (3.2) can be written in the form $\hat{h}_t = \mathsf{C}\mathsf{A}^{t-1}\mathsf{B}$ for $t > 0$. If we choose $\mathsf{A} = \mathrm{diag}(\lambda)$ then the impulse response becomes

$$\hat{h}_t = \sum_{n=1}^{d} R_n \lambda_n^{t-1} = \mathsf{C}[\mathrm{diag}(\lambda)]^{t-1}\mathsf{B} = \sum_{n=1}^{d} \mathsf{C}_n \mathsf{B}_n \lambda_n^{t-1}$$

The choice $\mathsf{B}_n = 1$ for all $n = 1, \ldots, d$, $\mathsf{B} = \mathbb{1}_d$ and $\mathsf{C}_n = R_n$ finalizes a modal canonical state-space realization of the distilled filter. The $\mathcal{O}(d)$ time complexity of the corresponding recurrent step is guaranteed by the decoupling of each state equation from another,

$$x_{t+1}^n = \lambda_n x_t^n + u_t \quad n = 1, \ldots, d$$

$$y_t = \sum_{n=1}^{d} R_n x_t^n + h_0 u_t.$$

Each of the $d$ state equations can be computed (in parallel) in $\mathcal{O}(1)$ time. The output equation is a dot product requiring $d$ multiplications and $d$ additions, hence the $\mathcal{O}(d)$ time compexity of the recurrence.

### C.8 Proof of Proposition 3.2

$x_T = (v_T, \ldots, v_{T-d})$ where $v = g * u$ and $g$ is the filter whose transfer function is $1/\mathrm{den}(\hat{H})(z)$ and can be evaluated in $\tilde{\mathcal{O}}(T)$.

Without loss of generality, let us assume to have converted the distilled filter in canonical form (i.e. we have unrestricted access to the coefficients of the rational transfer function) and let $\mathsf{D} = 0$. We use the notation of Section A.5. In $z$-domain, the state-to-input relation is given by

$$Y(z) = \mathsf{C}X(z) = [\beta_1 \quad \cdots \quad \beta_d]\, X(z)$$

On the other hand $Y(z) = \hat{H}(z)U(z) = q(z)/p(z)U(z)$. Therefore,

$$[\beta_1 \quad \cdots \quad \beta_d]\, X(z) = \frac{q(z)}{p(z)}U(z)$$

$$\Leftrightarrow \quad [\beta_1 \quad \cdots \quad \beta_d]\, X(z) = [\beta_1 \quad \cdots \quad \beta_d] \begin{bmatrix} z^{-1} \\ \vdots \\ z^{-d} \end{bmatrix} \frac{1}{p(z)}U(z)$$

$$\Leftrightarrow \quad X(z) = \begin{bmatrix} z^{-1} \\ \vdots \\ z^{-d} \end{bmatrix} \frac{1}{p(z)}U(z)$$

Let $V(z) = U(z)/p(z)$. From the shift property of the $z$-transform it holds,

$$\mathcal{Z}\{x\}(z) = X(z) = \begin{bmatrix} z^{-1} \\ \vdots \\ z^{-d} \end{bmatrix} V(z) \quad \Leftrightarrow \quad x_t = (v_{t-1}, v_{t-2}, \cdots, v_{t-d}) \ \forall t > 0.$$

$v$ can be obtained in $\tilde{\mathcal{O}}(L)$ time via an FFT-convolution of the input $u$ and $g$, the filter resulting from inverse transforming $1/p(z)$. The proof is convoluded setting $t = L$

## C.9 Proof of Theorem 4.1

**Notation.** We will be denoting the all $1$ row vector of size $k$, given by $[1 \quad 1 \quad \ldots \quad 1 \quad 1]$, and the all $0$ row vector of size $k$, given by $[0 \quad 0 \quad \ldots \quad 0 \quad 0]$, as $\mathbf{1}^k$ and $\mathbf{0}^k$, respectively. We will also construe the standard basis vector $e_i$ as a column vector in these notes. Next, we will adhere to the following matrix indexing convention: $\mathsf{A}_{ij}$ is the entry in the $i$th row and the $j$th column, $\mathsf{A}[i,:] \in \mathbb{F}^{1 \times n}$ denotes the $i$th row, and $\mathsf{A}[:,j] \in \mathbb{F}^{m \times 1}$ denotes the $j$th column of $\mathsf{A} \in \mathbb{F}^{m \times n}$. Here, we also use $\mathbb{0}^{m \times n} \in \mathbb{R}^{m \times n}$ and $\mathsf{I}_n$ to denote the matrix of all zeros and the identity matrix of dimension $n$, respectively. Moreover, we extend the outer product between two vectors to a tensor product using the symbol $\otimes$, the computation of which is carried out batch-wise with some dimension of one or both of the input tensors. Finally, we express the binary encoding of $i \in [n]$ in a row vector form, given by $\mathsf{B}_i \in \mathbb{Z}_2^{P_n}$, where $P_n$ is the closest power of 2 to $n$.

**Language and Model Description.** The language $\Lambda$ has $s$ keys and $s$ values: $L_K := \{k_1, \ldots, k_s\}$, $L_V := \{v_1, \ldots, v_s\}$. Formally, the language $\Lambda$ consists of sequences $x \in (L_K \times L_V)^s \times L_K$, where there is an associated mapping $f_x : L_K \to L_V$. For each sequence, the odd indices in $[L]$ belong to $L_K$, for $x_1, x_3, \ldots, x_L$, and we define

$$x_{2 \cdot i} = f_x(x_{2 \cdot i - 1}) \tag{C.3}$$

The last item $x_L \in \{x_1, x_3, \ldots, x_{L-1}\}$, called the *query*, must be one of the keys that has appeared in $x$ already. Our goal is to produce $f_x(x_L)$ at the end of the sequence, which we refer as the *associated value*. This problem is termed as the *associative recall problem* [55].

We will now outline the Hyena layer [2] with multiple heads as follows.

---

**Algorithm 1** Hyena

---

**Require:** Input sequence $u \in \mathbb{R}^{L \times D}$ from the previous layer, long convolution filter $\mathsf{T}_h$, number of heads $M$.

1: $q^m, k^m, v^m \leftarrow \mathsf{Projection}(u)$ for $m \in [M]$.
2: **for** $m = 1, \ldots, M$ **do**
3:     Perform the outer product $z^m \leftarrow k^m \otimes v^m \in \mathbb{R}^{L \times N \times N}$, where $N := D/M$.
4:     Apply the convolution independently and compute $y_t^m \leftarrow \mathsf{T}_h(z_t^m)q_t^m \in \mathbb{R}^{L \times N}$
5: Average the output $\overline{y} \leftarrow \left(\sum_m\right) y^m / M$
6: Retrieve the value $f(k_L)$ of the key $k_L$ from $\overline{y}[L,:]$.

---

In order to prove Theorem 4.1, we need the following technical statement concerning sparse recovery of a heavy-hitter.

**Proposition C.1** (Heavy-Hitter Recovery)**.** *Let $x \in \mathbb{R}^s$ be a vector with one entry bounded by $1 \pm \frac{1}{3\sqrt[4]{s}}$—referred as the* heavy-hitter—*and the rest of the entries bounded by $\pm\frac{1}{3\sqrt[4]{s}}$. Then, there exists a matrix $\mathsf{S}^{(m)} \in \mathbb{R}^{s \times O(\sqrt{s}\log s)}$ such that the position of the heavy-hitter in $x$ can be inferred from the average of $M$ measurements with $\mathsf{S}^{(m)}$ given by $\left(\sum_m x\mathsf{S}^{(m)}\right)/M$ with probability of error $\leq \frac{1}{s}$.*

Before presenting the proof of Proposition C.1, we use it to prove Theorem 4.1 as follows.

*Proof of Theorem 4.1.* We take $D = O(\sqrt{s}\log^2 s)$ and $M = 243 \cdot \log s$ so that $N = O(\sqrt{s}\log s)$ and use the same projections and filters for each head. We will start by describing the projections of the input. To this end, let $E : [L] \to 2s$ define a map from the row indices of $u$ to the keys $k_i$ and values $f_x(k_i)$ given by

$$E(t) = \begin{cases} i, & t \text{ odd}, \ x_t = k_i, \\ i + s, & t \text{ even}, \ x_{t-1} = k_i, \end{cases} \tag{C.4}$$

Here, we note that we also have

$$E(t) = E(t-1) + s, \quad t \text{ even} \tag{C.5}$$

as the even indices are defined as $x_t = f_x(x_{t-1})$ for $t$ even (C.3), whence $x_{t-1} \in L_K$ as $t-1$ is odd. Next, we can separate the keys $q$, queries $q$ and values $v$ from the input sequence $u$. For keys and queries, we will be using the Johnson-Lindenstrauss embedding [56]. We state its guarantee here.

> For a set of points $P \subseteq \mathbb{R}^s$, let $\epsilon, \delta > 0$ with $k \geq 2\ln\left(\frac{2s}{\delta}\right)/\epsilon^2$, and $f : \mathbb{R}^s \to \mathbb{R}^k$ be the randomly constructed linear map from [56], then with probability of error $\leq \delta$, we have
>
> $$|\langle f(x), f(y)\rangle - \langle x, y\rangle| \leq \epsilon$$

for all $x, y \in P$.

More precisely, we take $\mathsf{R} \in \mathbb{R}^{O(\sqrt{s}\log s) \times s}$ to be the matrix representation of $f$ with $\epsilon := \frac{1}{3\sqrt[4]{s}}$ and $\delta := \frac{1}{s^c}$ for some $c > 1$ so that $\mathsf{R}[:, i] = f(e_i)$. Thus, we define

$$q^m[t, :] = \begin{cases} \mathsf{R}[:, E(t-1)], & E(t-1) \le s \\ 0, & \text{otherwise.} \end{cases} \tag{C.6}$$

For values, we use the heavy-hitter recovery matrix as described in Proposition C.1 so that we have

$$v^m[t, :] = \begin{cases} \mathsf{S}^{(m)}[E(t), :], & E(t) > s \\ 0, & \text{otherwise,} \end{cases} \tag{C.7}$$

Further, using 1DConv (equivalently, in terms of polynomials, $h(X) := X$), we can shift the queries to get the projection for keys $k$ so that we have $k^m[t, :] = q^m[t-1, :]$

The Hyena filters, along with the specific convolution being performed by $\mathsf{T}_h$, are specifically described in terms of polynomial multiplications, for all $m \in 1, \dots, M$, as follows.

$$\mathsf{T}_h(X) := \sum_{i=0}^{L} X^i.$$

Here, we note that $\mathsf{T}_h(u)$ takes the cumulative sum over the input. That is, for all $i$, we have

$$\mathsf{T}_h(u)[i, :] = \sum_{j=0}^{i} u[j, :]$$

We will now compute $z^m$ as follows

$$z^m = k^m \otimes v^m$$

Further, applying the convolution, we get

$$\mathsf{T}_h(z_t^m) = \sum_{i=0}^{t} k^m[i, :] \otimes v^m[i, :]$$

For inference, it suffices to show that the last row of the output $y$ recovers the output with high probability. Indeed, let $t' \in [L]$ denote the row index of the value associated to the query such that the corresponding key has the following relation

$$u_{t'-1} = u_L. \tag{C.8}$$

Finally, we multiply by the query $q$ across $L$. Specifically, we now look at the computation of the $L$th row of $y$:

$$y^m[L, :] = \left( \sum_{t=0}^{L} k^m[t, :] \otimes v^m[t, :] \right) q^m[t, :]$$

$$= \sum_{t=0}^{L} \left( q^m[L, :]^\top k^m[t, :] \right) v^m[t, :]$$

$$= \sum_{t=0}^{L} \left( q^m[t'-1, :]^\top q^m[t-1, :] \right) v^m[t, :] \tag{C.9}$$

$$= \sum_{\substack{t \in [L] \\ t \text{ even}}} \left( (\mathsf{R}[:, E(t'-1)])^\top \mathsf{R}[:, E(t-1)] \right) \mathsf{S}^{(m)}[E(t), :] \tag{C.10}$$

$$= \sum_{\substack{t \in [L] \\ t \text{ even}}}^{L} \left( \mathsf{Re}_{E(t'-1)}^\top \mathsf{Re}_{E(t-1)} \right) \mathsf{S}^{(m)}[E(t), :] \tag{C.11}$$

Here, we are using the fact that $k^m[t'-1, :] = q^m[L, :]$ due to (C.8) in (C.9). We then change the indexing from (C.9) and (C.11) by observing that all the odd entries corresponding to values are zeroed out in $\mathbf{K}$ (cf. C.6). Finally, we simply substitute (C.6) and (C.7) in (C.10) and (C.11), respectively. Next, we define $x \in \mathbb{R}^{s+1}$ with

$$x_j := \mathsf{Re}_{E(t'-1)}^\top \mathsf{Re}_j \tag{C.12}$$

where $j = t - 1$ with $t \in [L]$ and $t$ even. Here, $x \in \mathbb{R}^{n+1}$ is a vector of size $s + 1$ as there are $s + 1$ such even numbers in $[L]$. Note that $x$ is the vector with a heavy-hitter from Proposition C.1. To see

this, observe that we have $\left|x_{E(t'-1)} - 1\right| \leq \frac{1}{3\sqrt[4]{s}}$ and $|x_j| \leq \frac{1}{3\sqrt[4]{s}}$ for all $j \neq E(t'-1)$. Using (C.11), with probability $\geq 1 - \frac{1}{s^c}$, we then have

$$y^m[L, :] = xS^{(m)}. \tag{C.13}$$

By Proposition C.1, we can then infer the position of the key at $t'$ with probability of error $\frac{1}{s}$. By the union bound, we can then retrieve the corresponding value with probability at least $1 - (\frac{1}{s} + \frac{1}{s^c})$. $\qquad\square$

We will now prove C.1 as follows.

*Proof of C.1.* We will assume that $s$ is a power of 2 for the sake of simplicity. We first specify how we will construct such an $S^{(m)} \in \mathbb{R}^{s \times O(\sqrt{s}\log s)}$. Let $h : [s] \to [\sqrt{s}]$ be a hash function. We define $\tilde{S} \in \mathbb{R}^{s \times \sqrt{s}}$ to be

$$\tilde{S}[:, i] = \sum_{j : h(j) = i} e_j.$$

That is, each column $i$ of $\tilde{S}$ is the sum of the standard basis vectors $\mathbf{e}_j$ such that $j$ is mapped by $h$ to $i$. In other words, the locations of the non-zero entries in column $i$ correspond to the preimage of $i$ under $h$. We then multiply each non-zero entry of $\tilde{S}$ independently at random by $\pm 1$. Next, we replace the $k$th row in $\tilde{S}$ by multiplying all non-zero $\pm 1$ entries at index $i$ with the binary representation of $i$ to get a matrix $S^{(m)} \in \mathbb{R}^{s \times (\sqrt{s} \times \log s)}$. That is, for a non-zero entry at index $i$ in row $k$, we replace the $i$th entry with $\pm 1 \cdot \mathbf{B}_i$. Note here that each column still has at most $\sqrt{s}$ non-zero entries. Finally, we stack $243 \cdot \log s$-many copies of $S^{(m)}$ as heads so that each copy produces independent measurements $xS^{(m)}$. Here, we want to emphasize that each such copy uses fresh randomness for multiplying the non-zero entry of $\tilde{S}$ independently at random by $\pm 1$.

Now, we will show that the average of the measurements with matrices $S^{(m)} \in \mathbb{R}^{s \times \sqrt{s}\log s}$ can locate the heavy-hitter in $x$, where $x$ is the vector of inner products from (C.12). For this purpose, we first specify the algorithm for decoding the heavy-hitter.

---

**Algorithm 2** Decoder

---

**Require:** The vector $y$ such that $y = xS$.
1: Split $y$ into $243 \cdot \log s$ blocks $y^{(m)} \in \mathbb{R}^{\sqrt{s}\log s}$, each of which is a result of multiplying $x$ by $S^{(m)}, m \in [243 \cdot \log n]$.
2: Take the average $\bar{y} \leftarrow \frac{1}{243 \cdot \log s} \sum_k y^{(m)}$.
3: $\boldsymbol{b} \leftarrow I_r(|\bar{y}|) \in \mathbb{R}^{\sqrt{s}\log s}$, cf. (C.14).
4: Retrieve $\boldsymbol{b}$ by isolating the binary representation of the position of the heavy-hitter in $x$.

---

Here, we define the function $I_r : \mathbb{R}^{\sqrt{s}\log s} \to \mathbb{Z}_2^{\sqrt{s}\log s}$ to $[xS]_m$ that rounds each entry of its input to the nearest integer:

$$I_r([xS]_m) = S^{(m)}[i, :]. \tag{C.14}$$

That is, in both cases, we retrieve the row in $S^{(m)}$ that corresponds to the heavy-hitter in $x$. Since the rows in $S^{(m)}$ are distinct, we can also infer the position of the heavy-hitter in $x$ with probability 1. We now show that $y = xS$, which consists of $243 \cdot \log n$ many independent copies of $y^{(m)} = x\hat{S}^{(m)}$. Instead, notice that we can analyze $\tilde{y} = x\tilde{S}$ since each $\hat{y}$ is the replacement of non-zero entries of $\tilde{S}$ with the binary representation of their indices times $y$. We will drop the superscript for now to avoid cluttering the notation. We can then make the following claim:

$$|\tilde{y}_i| = 1 \pm O(\epsilon \cdot \sqrt[4]{s}) \text{ and, for } j \neq i, |\tilde{y}_j| = O(\epsilon \cdot \sqrt[4]{s}). \tag{C.15}$$

For the above claim, we note that the first part follows from the latter as it suffices to show that all the non-zero heavy-hitters contribute $O(\epsilon\sqrt[4]{s})$ to the sum $\tilde{y}_i = \langle x, \tilde{S}[:, i]\rangle$. Since each column in $\tilde{S}$ only interacts with $\sqrt{s}$ sized sub-vector of $x$, each $\tilde{y}_j$ for non-heavy hitters can be expressed as

$$\tilde{y}_j = \langle \bar{x}, \bar{S}_j \rangle \text{ with } \bar{x}, \bar{S}_j \in \mathbb{R}^{\sqrt{s}},$$

where $\bar{S}_j \in \mathbb{R}^{\sqrt{s}}$ contains the non-zero entries of $\tilde{S}_j$ and $\bar{x}$ is obtained by extracting the entries with corresponding indices from $x$. Here, we have $\|\bar{x}\|_2 \leq \epsilon\sqrt[4]{s}$ since each entry associated with the non-heavy hitter is bounded as $x_i \leq \epsilon$, and thus, $\|\bar{x}\|_2 = \sqrt{\sum_i x_i^2} \leq \sqrt{\sum_i \epsilon^2} = \sqrt{\sqrt{s} \cdot \epsilon^2} = \epsilon\sqrt[4]{s}$. Consequently, as $\tilde{S}_j$ is independently random $\pm 1$, we then must have

$$\left|\langle \bar{x}, \bar{S}_j \rangle\right| \leq \frac{1}{3} \tag{C.16}$$

with constant probability for $j \neq i$. To see this, note that
$$\mathbb{E}[\langle \overline{x}, \overline{\mathsf{S}}_j \rangle^2] = \sum_{k,\ell} \mathbb{E}[\overline{\mathsf{S}}_{jk} \cdot \overline{\mathsf{S}}_{j\ell}] \cdot \overline{x}_i \cdot \overline{x}_j$$
$$= \|\overline{x}\|_2^2,$$
where the last equality follows since $\mathbb{E}[\mathsf{S}_{jk} \cdot \mathsf{S}_{j\ell}] = \delta_{k,\ell}$ by the distribution on entries of $\mathsf{S}_j$. Now, we use Jensen's inequality [57] to get the following bound on the expectation of $\left| \langle \overline{x}, \overline{\mathsf{S}}_j \rangle \right|$.
$$\mathbb{E}\left[ \left| \langle \overline{x}, \overline{\mathsf{S}}_j \rangle \right| \right] \leq \sqrt{\mathbb{E}[\langle \overline{x}, \overline{\mathsf{S}}_j \rangle^2]} \leq \epsilon \sqrt[4]{s}. \tag{C.17}$$
We then use the expectation above to bound the relevant probability as follows:
$$\Pr\left[ \left| \langle \overline{x}, \overline{\mathsf{S}}_j \rangle \right| \leq \frac{1}{3} \right] \geq 1 - \Pr\left[ \left| \langle \overline{x}, \overline{\mathsf{S}}_j \rangle \right| \geq \frac{1}{3} \right]$$
$$\geq 1 - 3\mathbb{E}\left[ \left| \langle \overline{x}, \overline{\mathsf{S}}_j \rangle \right| \right] \tag{C.18}$$
$$\geq 1 - 3\epsilon \cdot \sqrt[4]{s},$$
where we apply Markov's inequality [57] in (C.18). That is, we have shown that $\tilde{y}_j$ is bounded by $1/3$ with constant probability for $j \neq i$, and $\tilde{y}_i$ is thus bounded by $1 \pm \frac{1}{3}$. Note here that each of the $m$-copies $\tilde{y}_i^{(m)}$ will have identical guarantees.

Now, define the average $\overline{\overline{y}}_j := \frac{1}{243 \cdot \log s} \sum_m \tilde{y}^{(m)}$ so that $\overline{y}_j$ (line 3 in Decoder) is the corresponding replacement of the non-zero entries with the binary representation of their indices. We now claim that this average $\overline{\overline{y}}_j \leq 4/9 < 1/2$ with high probability for $j \neq i$. To this end, we employ the multiplicative Chernoff bound [57] on the independent random variables $\{\tilde{y}_j^{(h)}\}_h [0,1]$ with $\mathbb{E}\left[ \sum_m \tilde{y}_j^{(m)} \right] \leq 81 \cdot \log s$ to get
$$\Pr\left[ \overline{\overline{y}}_j > \frac{4}{9} \right] = \Pr\left[ \sum_m \tilde{y}_j^{(m)} > \left( 1 + \frac{1}{3} \right) \frac{1}{3} \cdot 243 \cdot \log s \right]$$
$$\leq \Pr\left[ \sum_m \tilde{y}_j^{(m)} \geq \left( 1 + \frac{1}{3} \right) 81 \cdot \log s \right],$$
$$\leq \exp\left( - \left( \frac{1}{3^2} \cdot 81 \cdot \log s \right) / 3 \right),$$
$$= \frac{1}{s^3}.$$
Therefore, we have shown that the average $\overline{\overline{y}}_j$ is less than $1/2$ with probability at least $1 - \frac{1}{s^3}$ for $j \neq i$. Consequently, we will have $\overline{\overline{y}}_i$ bounded by $1 \pm \frac{1}{2}$. Using the union bound over each $j \neq i$ and the $\log s$ bits in the binary representation of $j$, we can then show that $\overline{y}_{j+m} < 1/2$ for each $j \neq i, m \in [0, \log s]$ with probability $1 - \frac{\log s}{s^2} \gg 1 - \frac{1}{s}$. $\qquad \square$

# D    Experimental Details

## D.1    Pre-training

To verify the effect of introducing heads to Hyena as described in Section 4, we train a series of models on THE PILE [11]. All MultiHyena models are set to 8 heads, and otherwise use the same hyperparameters of Hyena models of equivalent size. We set weight decay of Hyena filter parameters to 0, and lower the frequency of sine activations in the implicit MLP to 4. We follow the setup of [2], and first train models for 5, 10 and 15 billion tokens, adjusting the learning rate scheduler accordingly. Then, we train for 300 billion tokens. The results are reported in Tables 5.1 and 5.1.

## D.2    Distillation Analysis

Distilling pre-trained long convolution sequence models (LCSM) with LaughingHyena can introduce errors on the convolution filter, which then propagate to the outputs.

**Setup**    We perform a series of extensive experiments on all variants of LCSM, including pre-trained H3 models of sizes 125 million, 355 million, 1.3 billion and 2.7 billion parameters; Hyena of size 153 million parameters, and MultiHyena of size 153 million parameters. For H3 models, we report approximation errors on both shift as well as diagonal SSMs (reported as IIR and FIR). Each point corresponds to distillation carried out at a particular order, using LaughingHyena modal interpolation. To optimize the parameters of the modal form, we use gradient-based optimization and minimize the $\ell_2$ discrepancy between filters in time domain. In particular, we use the ADAMW [58] optimizer with learning rate $3 \cdot 10^{-4}$, and a cosine annealing decay schedule down to $10^{-6}$ after 30 thousand iterations. Each individual filter of every layer is distilled in the same way.

**Discussion**    The errors are shown in Figures D.1, D.1, D.2, D.3, D.4 and D.5. We observe H3 filters to be easier to distill into recurrences with small state without introducing significant errors, whereas Hyena variants learn filters with larger effective dimensions. This provides further evidence that training with implicit convolutions may yield in general more expressive filters.

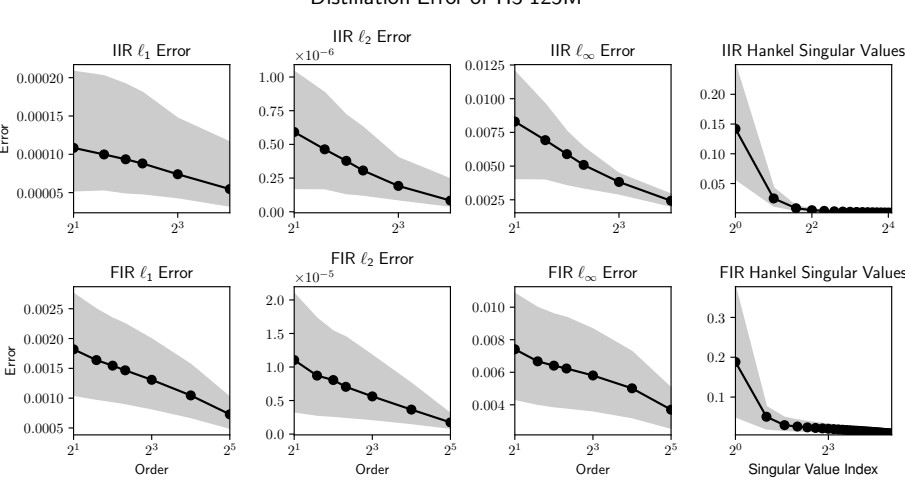

Figure D.1: Mean, lower and upper bounds across channels and layers of the distillation errors on 125M H3 model for both its IIR and FIR filters.

### D.2.1    Pretrained Filters: Effective Dimension

**Visualizations**    We qualitatively investigate LCSM filters at initialization and after pretraining. This visual inspection (Figures D.6, D.7 and D.8) complements the distillation error analysis of Section D.2.

**Distribution of Hankel singular values**    We further compute the distribution of Hankel singular values of each long convolution filter in different models. The decay in the spectrum quantifies how *easy* it is to find a compact modal form with LaughingHyena, and serves as a proxy measure of effective dimension of the convolution. The results are shown in Figures D.9 and D.10.

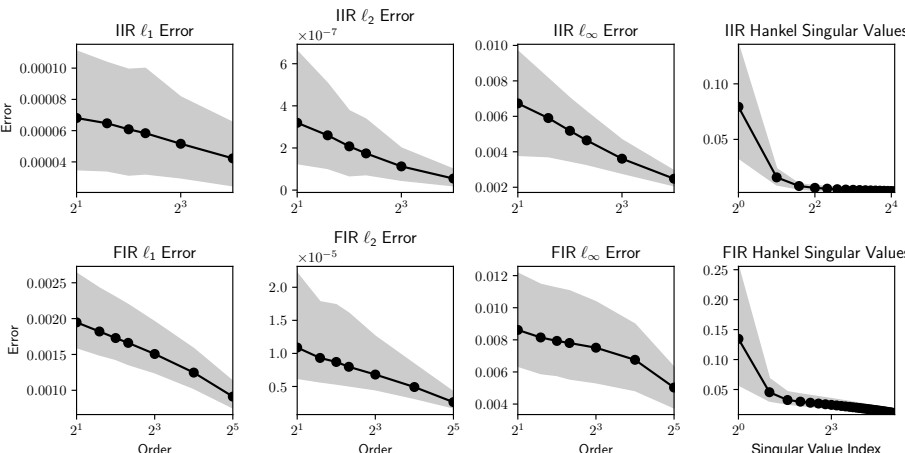

Figure D.2: Mean, lower and upper bounds across channels and layers of the distillation errors on 355M H3 model for both its IIR and FIR filters.

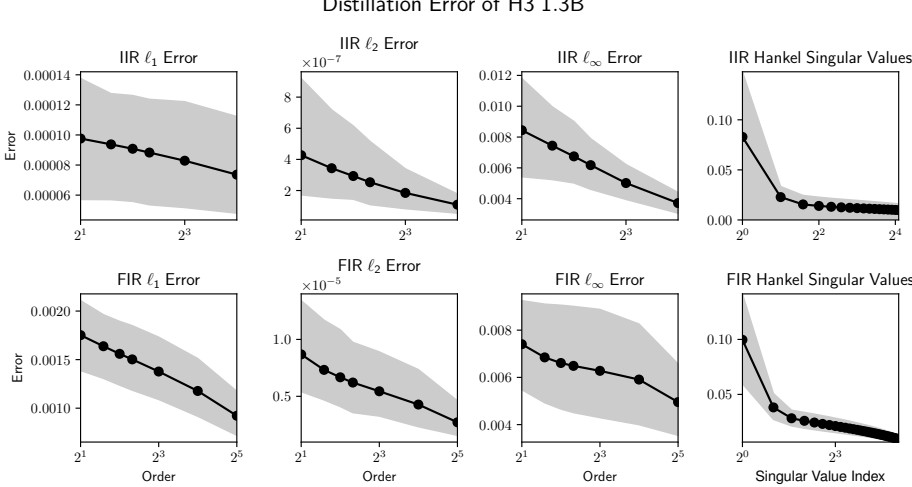

Figure D.3: Mean, lower and upper bounds across channels and layers of the distillation errors on 1.3B H3 model for both its IIR and FIR filters.

### D.3  Downstream Evaluation

We benchmark the downstream performance of MultiHyena and distilled MultiHyena on standard language modeling tasks from the LM-Eval-Harness [42] and HELM [41] suites. As a reference baseline, we evaluate Pythia [44] 160M.

Our objective is to quantify the absolute performance of MultiHyena and the downstream impact of distillation. We use the same procedure outlined in Section D.2 to distill MultiHyena.

### D.4  Benchmarking

To demonstrate the superior performance of Laughing Hyena for autoregressive generation, we conduct a series of experiments to benchmark its latency, throughput, and memory usage for autoregressive generation with initial prompt length $T$ and number of generated tokens $K$. For each experiment, we compare the performance of Laughing Hyena against a Transformer, a hybrid H3-attention model with 2 attention layers and a Hyena model. The latter two have been shown to match or achieve lower perplexity than Transformers on standard datasets (WIKITEXT103 and THE PILE). All experiments are carried out on a NVIDIA A100 with 80GB in float16 precision. Missing measurements for any model indicate Out of Memory (OOM) errors while doing autoregressive inference for that particular model.

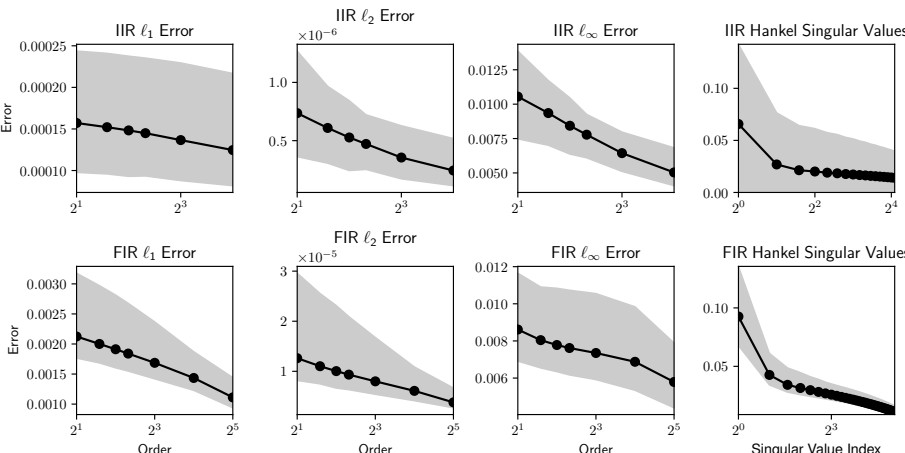

Figure D.4: Mean, lower and upper bounds across channels and layers of the distillation errors on 2.7B H3 model for both its IIR and FIR filters.

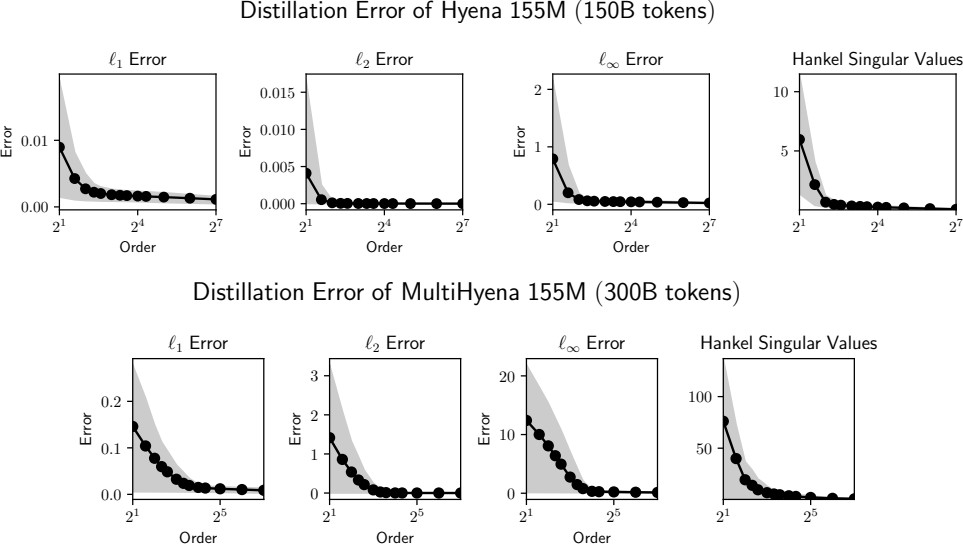

Figure D.5: Mean, lower and upper bounds across channels and layers of the distillation errors on Hyena and MultiHyena models.

**Peak throughput**  We first evaluate the throughput (number of tokens generated per second) across different batch sizes, using a typical generation workload consisting of a prompt of length 512 and generating 256 tokens. Figure 1.1 measures peak throughput of different models. Since Laughing Hyena does not require caching intermediate kv-projections during generation, reduced memory requirements at a fixed model size allow it to process larger batch sizes.

**Prompt length**  Autoregressive generation in Laughing Hyena is achieved through a two-step process: an initial prefill step that uses the length$-T$ prompt to initialize the state $x_T$ and that generates all $K$ tokens. In Figure 5.3 we demonstrate how the prefill step scales for different prompt lengths, keeping batch sizes fixed at 64. Since prefilling in Laughing Hyena is carried out efficiently via convolutions (as described in Section 3.4), throughput scales more favorably than Transformers. Other models capable of prefilling via convolutions also achieve higher throughputs than Transformers but are ultimately slower than Laughing Hyena during the generation phase.

**State throughput**  We measure the impact of SSM state dimension on the throughput of Laughing Hyena. Keeping batch sizes fixed reveals minimal impact for all dimensions smaller than

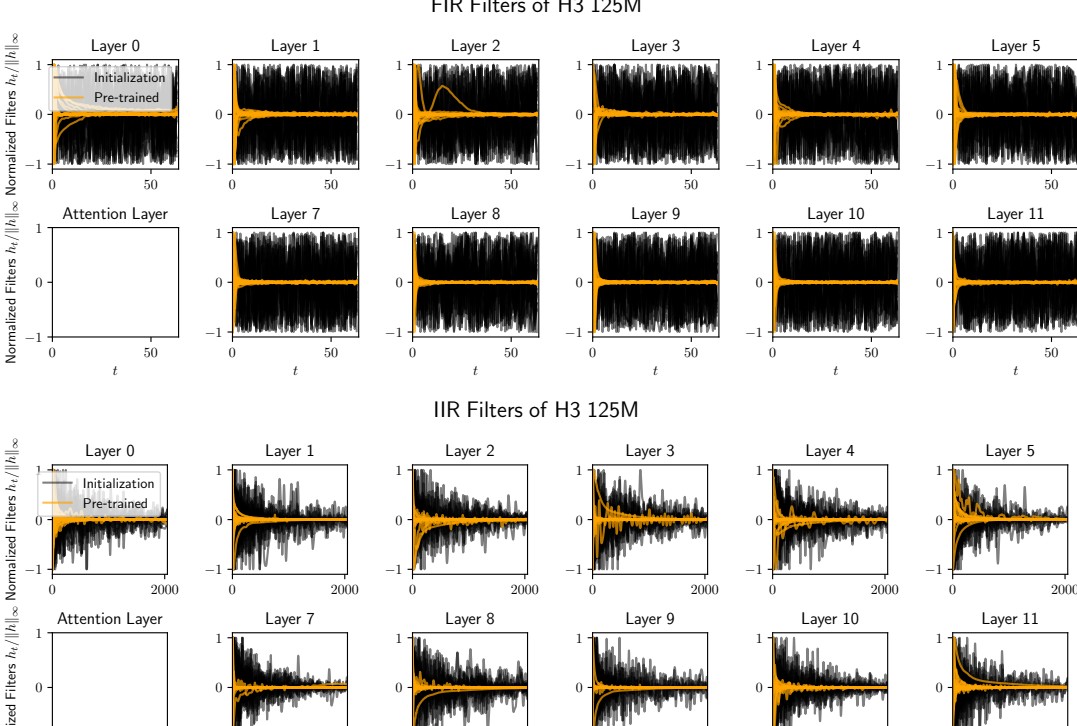

Figure D.6: Initialized and pre-trained convolution filters of H3.

100, which are sufficient to distill all models discussed in this work. All other measurements provided in this Section are carried out with a standard order 16. We note that it may be possible to further increase peak throughput by leveraging reduced memory footprints achieved by extremely small SSMs.

**Latency over sequence length**   We benchmark the time taken to generate a variable number of tokens, starting from a prompt of length 512 tokens at batch size 1 (Figure D.11). Laughing Hyena tracks highly optimized Transformers. We note that Laughing Hyena is asymptotically more efficient than Transformers; however, this regime is bottlenecked by hardware-specific implementation details and optimizations. We expect optimized, platform-specific implementations of Laughing Hyena to outperform Transformers even at batch size 1. When the prompt is long, the prefilling step becomes the bottleneck, and all convolutional models outperform Transformers.

**Parameter scaling**   To better understand how the performance of Laughing Hyena scales, we benchmark its latency, throughput, and peak memory utilization for autoregressive generation and 125M, 355M, 1.3B, 2.7B and 6.7B parameters. We compare the performance to that of Transformers, Hybrid-H3, and Hyena at the same number of parameters and report the results in Figure D.11. For the latency measurement, we use a batch size of 1 and benchmark the time taken to generate 128 tokens, starting from a prompt of length 512 tokens. For throughput and peak memory scaling against the number of parameters, we use a batch size of 64 and measure the throughput for generating 256 tokens starting with a prompt of length 512.

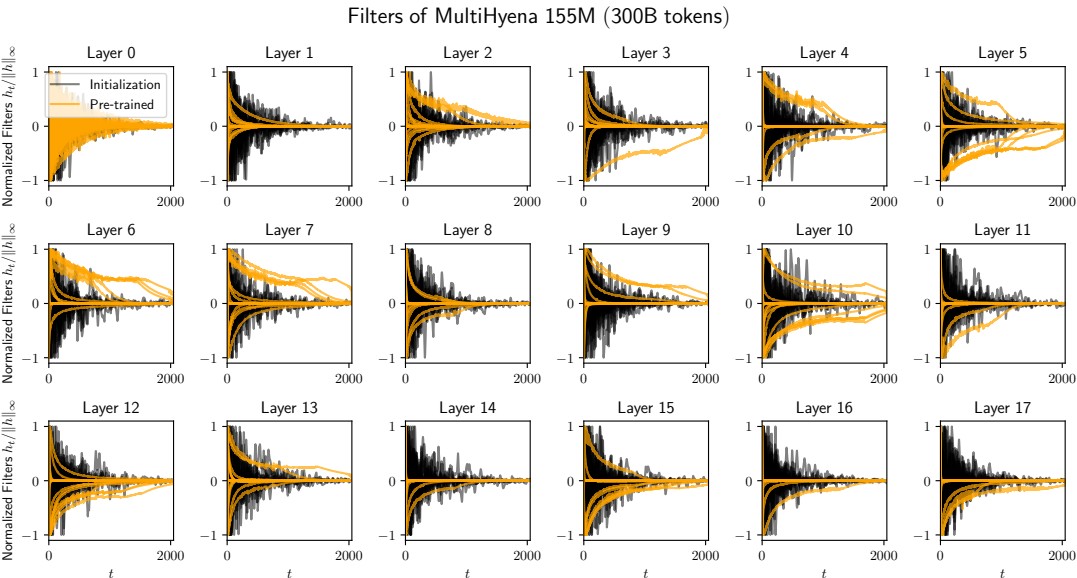

Figure D.7: Initialized and pre-trained long convolution filters of MultiHyena.

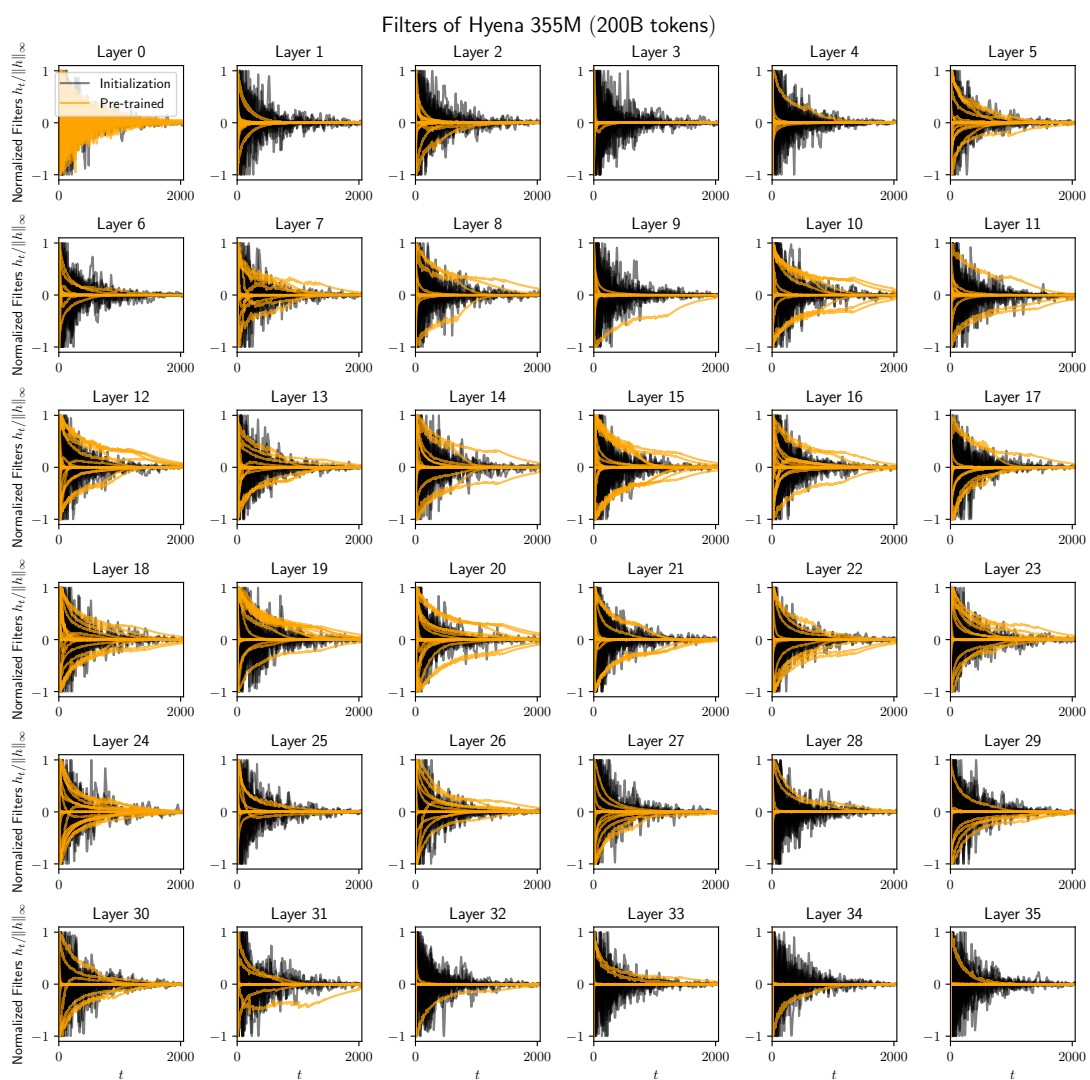

Figure D.8: Initialized and pre-trained long convolution filters of Hyena (355 $M$).

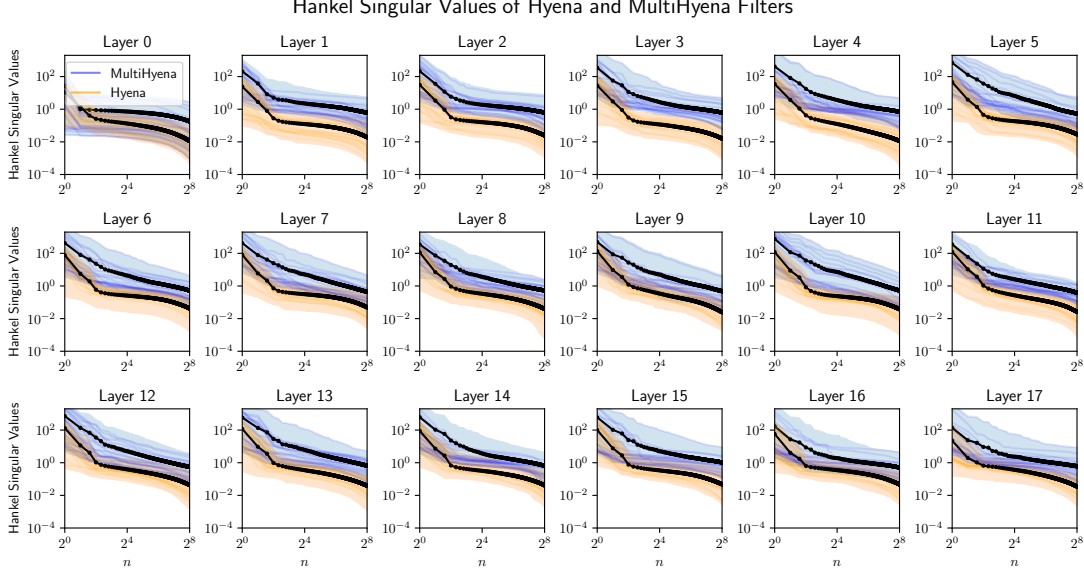

Figure D.9: Distribution of Hankel singular values for Hyena and MultiHyena long convolution filters. MultiHyena filters have larger effective dimension, as evidenced by slower decay.

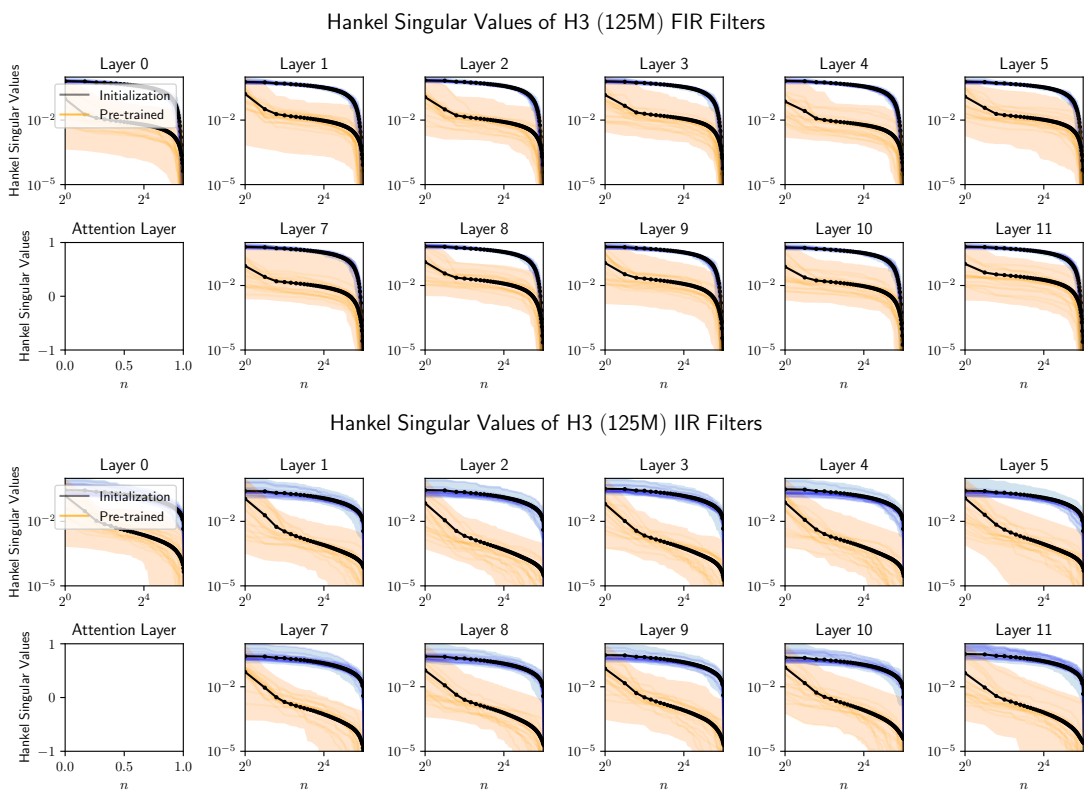

Figure D.10: Distribution of Hankel singular values for H3 long convolution filters. The values decay rapidly.

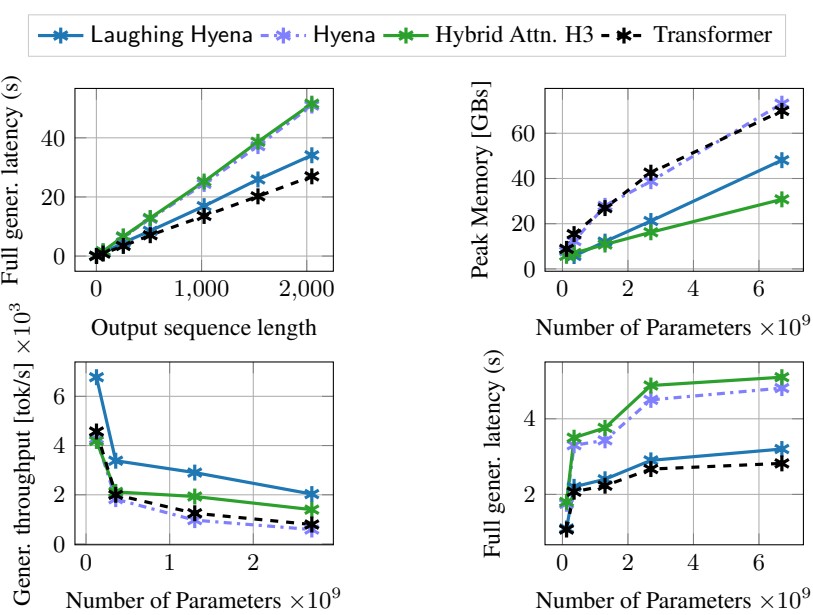

Figure D.11: Generation latency, throughput and peak memory of Transformers, H3, Hyena and Laughing Hyena.

# E Additional Experiments

## E.1 Associative Recall with MultiHyena

We follow the setup of [2] and train 2-layer Hyena and MultiHyena (with 8 heads) to solve associative recall via a standard next-token prediction objective. We focus on the sequence length 64k, high vocabulary size setting, and push vocabulary sizes past the maximum values considered in [2]. At vocabulary size 60, a difference between MultiHyena and Hyena can be observed (Table E.1), as experimental support for Theorem 4.1.

| Model | Accuracy |
|---|---|
| Hyena | 65 |
| MultiHyena | 98 |

Table E.1: Associative recall accuracy, sequence length 64k, vocabulary size 60.

## E.2 Analysis of Hankel Singular Values of Pretrained Large Convolution Sequence Models

## E.3 Model Order Reduction of H3

The H3 model is constructed with a combination of diagonal SSMs and shift SSMs. There exists various classical model order reduction techniques for these different types of SSMs. The following sections aim to present the formulation and effectiveness of two classical approaches on obtaining the compressed representation of a H3 model. More specifically, we study modal truncation and balanced truncation for compressing diagonal SSMs and shift SSMs respectively.

### E.3.1 Modal Truncation

A discrete diagonal SSM ($A = \text{diag}(\lambda_1, \ldots, \lambda_d)$, $B \in \mathbb{C}^{d \times 1}$, and $C \in \mathbb{C}^{1 \times d}$) can be directly converted into a residue-pole transfer function as follows:

$$(A, B, C) \rightarrow H(z) = \sum_{i=1}^{d} \frac{r_i}{z - \lambda_i}, \tag{E.1}$$

where residue $r_i = B_i C_i$. Modal truncation aims to compress such a transfer function by essentially reducing the summation over $d$ to $n < d$, of the $n$ most influential modes. The influence from each node can be isolated by expressing it using the $h_\infty$ norm of the system as follows:

$$\|H(z)\|_\infty = \sum_{i=1}^{d} \left\| \frac{r_i}{z - \lambda_i} \right\|_\infty \leq \sum_{i=1}^{d} \frac{|r_i|}{|1 - |\lambda_i||}. \tag{E.2}$$

Each mode $i$ can be ranked using the bound formulated above. Subsequently, the $d - n$ lowest modes could be discarded to form a reduced order model. Figure E.1 illustrates the monotonically decreasing $l_\infty$ error with the increase in system order. However, this model reduction approach is only suitable for diagonalizable SSMs.

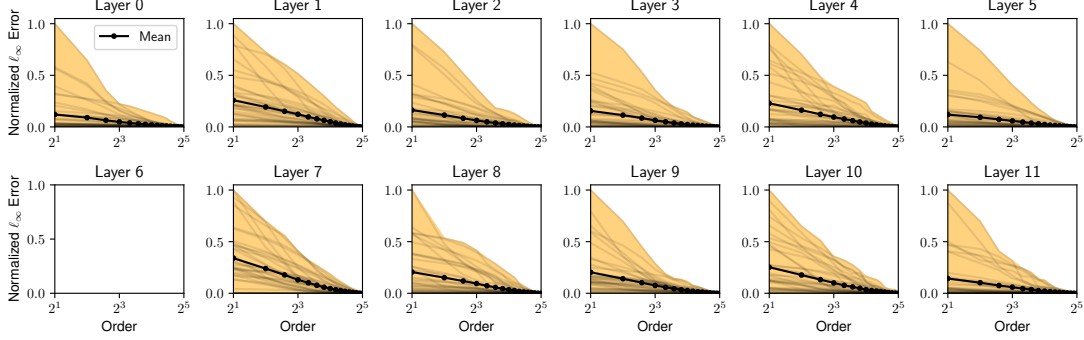

Figure E.1: Modal truncation model reduction error ($\|h(t) - h_n(t)\|_\infty$) across all diagonal SSM layers of the trained H3 125M model.

### E.3.2 Balanced Truncation

A balanced SSM realization is one in which the observability $P$ and controllability $Q$ gramians are equal and diagonal. Such a realization can be formulated with the following Lyapunov equations [24]:

$$\mathsf{A}\Sigma\mathsf{A}^\top + \mathsf{B}\mathsf{B}^\top = \Sigma,$$
$$\mathsf{A}^\top\Sigma\mathsf{A} + \mathsf{C}^\top\mathsf{C} = \Sigma,$$

(E.3)

where $P = Q = \Sigma = \mathrm{diag}(\sigma_1, \ldots, \sigma_d)$ and $\sigma_i \geq \sigma_{i+1}$.

Results from [59] shows that the $n-$order model reduction error is bounded by:

$$E_\infty \triangleq \|H(s) - H_n(s)\|_\infty \leq 2\sum_{i=d-n}^{d} \sigma_i.$$

(E.4)

Therefore, an $n-$order partition of the full balanced realization can be chosen, such that the discarded orders are the $d - n$ lowest contributor to the error. The steps taken by [24], computes the $n-$order partition of the balanced realization as follows:

1. Form a Hankel matrix $\mathsf{S}_d$ from the impulse response $h_{1:n}$ of the shift SSM.
2. Obtain the eigenvector matrix $V \in \mathbb{C}^{d\times d}$, and the eigenvalues $\lambda = \sigma^2$ via the eigen-decomposition of $\mathsf{S}_d$.
3. Choose the truncated model's order $n < d$, based on the bound in Equation E.4.
4. Compute the state-space matrices as follows:

$$\mathsf{A} = V_{2:d,1:n}^\top V_{1:d-1,1:n}, \quad \mathsf{B} = V_{1,1:n}, \quad \mathsf{C} = h_{1:d}^\top V_{1:d,1:n}, \quad \mathsf{D} = h_0.$$

(E.5)

This model reduction technique was applied to a trained 125 million parameter H3, MultiHyena, and Hyena models as shown in Figures E.2, E.3, and E.4 respectively. It could be noted that the all models encountered an undesirable non-monotonic error reduction with the increase in order. Moreover, order reduction configurations such as the one in Figure E.3 layer 15 display signs of numerical instability.

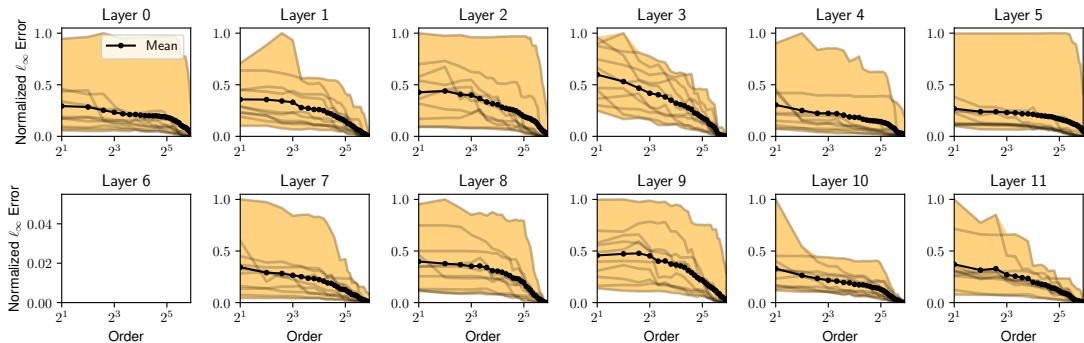

Figure E.2: Balanced truncation model reduction error ($\|h(t) - h_n(t)\|_\infty$) across all shift SSM layers of the trained H3 125M model. Note that Layer 6 is an Attention layer, therefore balanced truncation model order reduction is not possible.

Balanced Truncation Model Reduction Error of MultiHyena

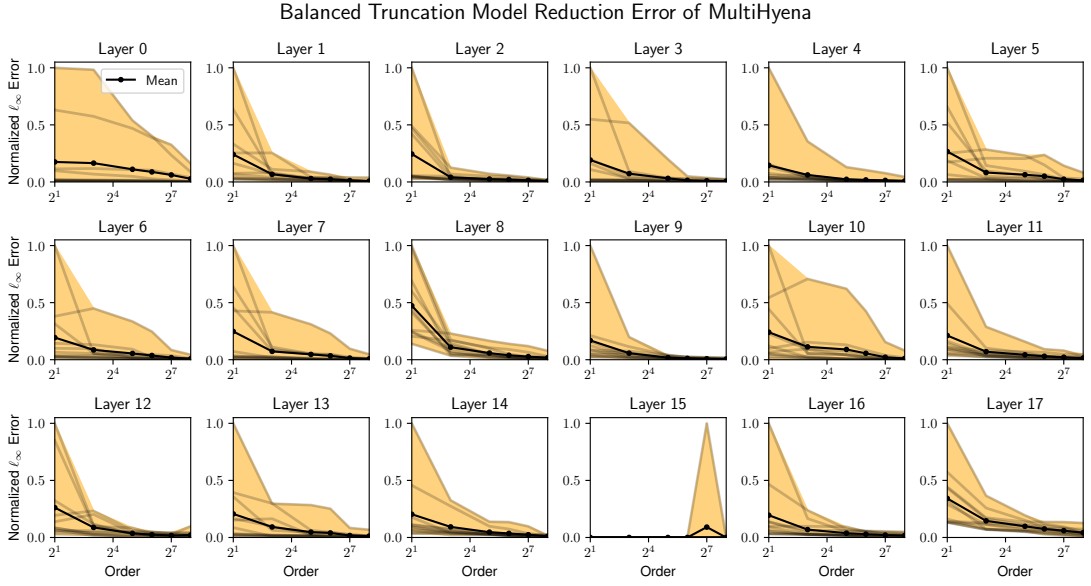

Figure E.3: Balanced truncation model reduction error ($||h(t) - h_n(t)||_\infty$) across all convolutional layers of the trained MultiHyena 155M model.

Balanced Truncation Model Reduction Error of Hyena

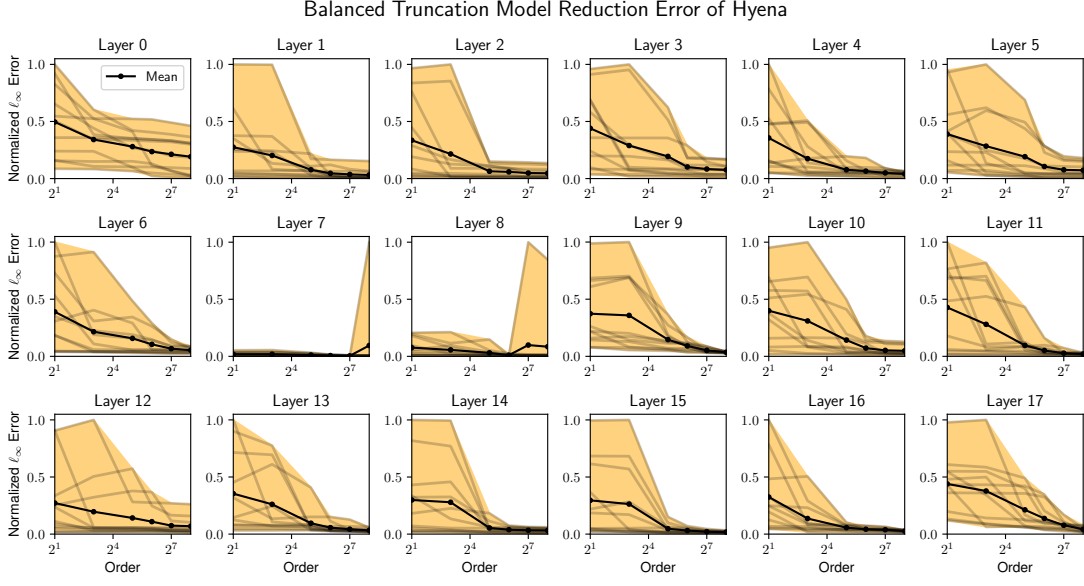

Figure E.4: Balanced truncation model reduction error ($||h(t) - h_n(t)||_\infty$) across all convolutional layers of the trained Hyena 155M model.

