# OpenReview forum: "Laughing Hyena Distillery: Extracting Compact Recurrences From Convolutions"
_NeurIPS.cc/2023/Conference — NeurIPS 2023 poster_

### Official Review · Reviewer_n6Pj · 2023-07-05

**Soundness:** 3 good
**Presentation:** 3 good
**Contribution:** 3 good
**Rating:** 6
**Confidence:** 4

**Summary:**

This article addresses two issues:

1. The low efficiency of the LonvConv-based model during inference.
2. Whether it is advantageous to perform independent long convolutions on each channel or reduce the total number of filters without loss in quality.

To tackle problem (1), the authors propose distilling the LonvConv into a Diagonal State space model and train it using the $\ell_2$ loss function.

For problem (2), the authors suggest sharing long convolution coefficients across multiple channels, resulting in the MultiHyena structure.

The effectiveness of the proposed methods is validated on multiple datasets.

**Strengths:**

Distilling LonvConv into a Diagonal State space model is indeed a novel and meaningful approach. The conclusion of sharing long convolution coefficients across multiple channels is also innovative.

**Weaknesses:**

I think the main issue with this article lies in the focus of the writing. From Equation 3.4, it is clear that the ultimate goal is to represent the LonvConv coefficients using a Diagonal State Space Model. However, a significant portion of the article is spent describing unrelated aspects. The most crucial part, determining the hidden dimension of the State Space Model, is merely brushed over, even though it is the key factor that affects the quality of the distillation and the efficiency of the final inference. On the other hand, the motivation behind the design of MultiHyena should be addressed in the main text since it is of utmost importance.

**Questions:**

1. The method for determining the hidden dimension of the State Space Model needs to be explained in more detail. I referred to Appendix E.3.2, and I'm wondering if the core idea is to perform an SVD decomposition and then select the dimension for dimensionality reduction based on the eigenvalues?

2. The solution to Equation 3.4 requires a more detailed algorithm description or pseudocode to help readers follow along. I attempted to replicate it following Appendix B.1, but the results were not quite good. Could you provide the training configurations and an estimate of the training time?

3. The motivation behind the design of MultiHyena should be included in the main text, and ablation studies (sharing coefficients vs not sharing coefficients) should be conducted to validate the design's rationale. The experiments should compare the effects and speeds.

4. Does MultiHyena utilize the Local conv + Global conv structure of Hyena? If so, how many layers are used? This should be mentioned in the experiments.

5. Is the Algorithm 1 on page 28 is the implementation of Figure 4.1?

6. Regarding the implementation of MultiHyena, in Algorithm 1 on page 28, for $z^m_t \in \mathbb R^{L\times N\times N}$, what does the subscript $t$ represent? On the other hand, is the shape of $T_h$ $L$ (all features share one set of convolution coefficients) or $L\times N\times N$ (each feature has independent convolution coefficients)?

7. Continuing with the implementation of MultiHyena, in Algorithm 1 on page 28, is the shape of $T_hz^m_t$ ${L\times N\times N}$? If so, does $T_h(z^m_t)  q_t^m$ mean performing matrix multiplication between $[T_h(z^m_t)]_i \in \mathbb R^{N\times N}$ (for $i=0,\ldots,L-1$) and $q_t^m\in \mathbb R^{N}$, resulting in an output of shape $\mathbb R^{N}$? If not, what is the computation like?

8. The algorithm's output is $\bar{y} \leftarrow\left(\sum_m\right) y^m / M\in \mathbb R^{L\times N}$, while the input has a shape of $L\times D$. Is this inconsistency problematic, or did the algorithm omit something?

9. Algorithm 1 has several ambiguities. It is suggested to reorganize it for better clarity.

**Limitations:**

Yes.

---

> ### Author Rebuttal · Authors · 2023-08-09
>
> We thank the reviewer for the detailed feedback and insightful questions. Please see the general response for additional details and experiments.
>
> > The method for determining the hidden dimension of the state space model needs to be explained in more detail. I referred to Appendix E 3.2, and I’m wondering if the core idea is to perform an SVD decomposition and then select the dimension for dimensionality reduction based on the eigenvalues.
>
> We thank the reviewer for the feedback. The core idea is indeed to perform an SVD decomposition of the Hankel operator constructed with the filter. We can then look at the decay of the singular values and select the dimension for distillation. Note that however we often select state dimensions well below the actual rank of the Hankel operator (Ho-Kalman [1]). Luckily, the AAK theory [2] gives us a bound on the approximation error (in terms of the Hankel norm) for a given state dimension. In particular the $d$th singular value is equal to the approximation error with the optimal $(d-1)$-dimensional state-space model. The eventual rationale for the choice of state dimension is however the performance of the distilled model in downstream tasks. Appendix E 3.2 mainly contains the details for classic model-order reduction methods (balance truncation) which we test as baseline distillation procedure on state-space models in the H3 architecture. You can also find further details in the reply to Reviewer wEPT.
>
> > The solution to Equation 3.4 requires a more detailed algorithm description or pseudocode to help readers follow along. I attempted to replicate it following Appendix B.1. but the results were not quite good. Could you provide training configurations and an estimate of the training time?
>
> We are happy to clarify and elaborate on the method. In [gist](https://gist.github.com/neurips2310098/8bb0869e9bb6f083df9bab240ea47376) ([checkpoint](https://drive.google.com/file/d/1tnLzWAlB4In0nUDsiT3w_w_rnL5MGhf6/view?usp=share_link)) we provide a self-contained notebook where we showcase the distillation procedure with our modal approximants for a small MultiHyena (125M). On a single RTX3090 GPU the distillation takes about one minute per layer with an order 16 state-space model. We are happy to provide further details on the training procedure and hyperparameters.
>
> > The motivation behind the design of MultiHyena should be included in the main text, and ablation studies (sharing coefficients vs not sharing) should be conducted to validate the design’s rationale. The experiment should compare effects and speeds.
>
> This is an important point. The design of MultiHyena, as mentioned in Section 4, is motivated by the observation that the effective dimensionality of long convolution filters in Hyena shrinks during training. Then, it can be advantageous (reducing the number of parameters) to share a smaller number of independent filters across "heads". Below, we provide an additional ablation that shows how sharing filters has a minimal impact on pretraining perplexity. Note that in the models below, we do not perform the multi-head computation outlined in Section 4; we only share filters in a vanilla Hyena model.
>
> | **Model (~155M)**      | **10B** |
> | ----------- | ----------- |
> | Hyena-4     | 11.8     |
> | Hyena-8   | 11.9      |
> | Hyena-16  | 11.9      |
> | Hyena-32  | 12.1      |
>
> When we combine weight-sharing with the proposed multi-head layer structure (which itself is motivated by Theorem 4.1), we obtain significantly lower perplexity:
>
> | **Model (~155M)**      | **10B** |
> | ----------- | ----------- |
> | MultiHyena-4     | 10.8     |
> | MultiHyena-8   | 10.4        |
> | MultiHyena-16  | 10.1      |
> | MultiHyena-32  | 9.6       |
>
>
> >  Does MultiHyena utilize the local convention + global convolution structure of Hyena? If so, how many layers are used? This should be mentioned in the experiments.
>
> At the scales we tested so far (125M, 355M, 1B), MultiHyena utilizes the same exact structure of Hyena, with the notable exception of having to parametrize only $D / H$ long convolutional filters.
>
> > Is the Algorithm 1 on page 28 is the implementation of Figure 4.1
>
> We will move it to the main text in the revised version of the paper to clarify the differences between Hyena and the proposed multi-head version.
>
> > Dimensions in MultiHyena
>
> We thank the reviewer for the chance to clarify. $T^m_h$ in this case is $L \times L $, and $q^m$ is $L \times N \times N$. The convolution is applied over the sequence length dimension ($T^m_h q^m$), in **parallel** for all the $N \times N$ "sequences" in $q^m$. The result is also a $N \times N \times L$, which is then collapsed to $N \times L$ via a dot product, and finally concatenated with the other $M$ chunks to produce $D \times L$.
>
> > Question on algorithm output dimensions
>
> In the standard model, we concatenate the $M$ chunks of size $N$ instead of averaging, yielding the desired $L \times D$ dimensions. The construction described in the comment is a special case (concatenation followed by a dense layer with specific weights) used for the proof of the Theorem.
>
> [1] Ho, B. L., and R. E. Kalman. "Effective construction of linear state-variable models from input/output functions." REGELUNGSTECHNIK (1966).
>
> [2] V.M. Adamjan, D.Z. Arov, M.C. Krein, "Analytic properties of Sc~nidt pairs for a Hankel operator and the generalized Schur-Takagi problem", Mat. USSR Sbornik, vol. 15, p. 31-73, 1971.
>
> [3] Orvieto, Antonio, et al. "Resurrecting recurrent neural networks for long sequences." arXiv preprint arXiv:2303.06349 (2023).

---

> > ### Author Response · Authors · 2023-08-20
> > **Response**
> >
> > Thank you again for the review and the positive comments! As the discussion period is about to end, we leave a gentle reminder to take into account our response (including the additional evaluations and modifications to improve clarity) into the final evaluation of the work.

---

### Official Review · Reviewer_6dqz · 2023-07-06

**Soundness:** 4 excellent
**Presentation:** 2 fair
**Contribution:** 3 good
**Rating:** 7
**Confidence:** 4

**Summary:**

The paper proposes distilling convolutional models for autoregressive sequence generation into recurrent (state-space) models. A key limitation of recently proposed convolutional models is that they use convolutional filters that extend potentially infinitely into the past, and the same techniques that yield efficient training do not transfer over to token-by-token generation. This paper proposes a post-training and data-free distillation step that replaces such convolutional models with a recurrent approach that uses constant time and memory at each step of generation during inference.

**Strengths:**

The key strength of the paper is that proposes a novel and theoretically grounded distillation method, which achieves good approximation bounds without being tied to specific data or involving what amounts to an additional round of training.

Also, convolutional and state space models as a whole are an underexplored area in recent work when compared to Transformers and attention, and this paper puts forward a novel method of linking the two, both of which contribute to the originality of the paper.

**Weaknesses:**

My biggest reservation based on my understanding of the paper is that I didn't get a qualitative sense of what might be lost as part of the distillation process. Is there some intuition of what is the worst-case qualitative behavior of a convolutional filter that can't be tightly approximated with LaughingHyena?

This question comes to mind because in the world of Transformers, it is no secret that most of the computational power spent on quadractic attention goes to waste. Just a small subset of the efficiency work there includes pruning entire heads, limiting each head to a head-specific attention context window, or even doing sliding-window attention with a fixed context length for the entire model. More in line with the present paper, work like Performer has developed an approximation for converting attention-based models into recurrent models -- at a cost. Notwithstanding the theory of the tightness of that last approximation, and equivalent performance of many to the Transformer that is demonstrated in some of the papers introducing these methods, there inevitably arises some situation where none of the approximations match the Transformer in quality. I worry that the present approach might fall into a similar pattern. A recurring theme in this area is that any method that sacrifices the ability to have long context, or to perform associative recall, is suspect. That's why when it comes to these approximations of convolutions, it would helpful to know whether the approximation is effectively some form of context-truncation in disguise, and if not what the qualitative cost is.

**Questions:**

How well does the LaughingHyena architecture perform on the associative recall task, especially as compared to Hyena (or MultiHyena)? Is the point beyond which the models fail to perform the task (in terms of sequence length or vocabulary size) different between the two?

For LM-Eval-Harness and HELM, have you tried a baseline of taking the impulse response from Hyena/MultiHyena and truncating it to a finite impulse response? A sliding window seems like one of the simplest approximations to try in the world of convolutions, and it would be helpful to know if maybe some defect of the tasks or the base model results in nothing more being required. This would, in fact, be a useful baseline to have in the paper.

**Limitations:**

The paper could be improved with a little bit of additional discussion regarding limitations of the distillation/approximation.

---

> ### Author Rebuttal · Authors · 2023-08-09
>
> We thank the reviewer for insightful comments and questions. We particularly appreciated the suggestion of a specific additional baseline (FIR truncation).
>
> > Is there some intuition of what is the worst-case qualitative behavior of a convolutional filter that can’t be tightly approximated with LaughingHyena?
>
> Any approximation error on the convolutional filters is propagated through the depth of the neural network to the logits (in the case of language). If such an error is above a certain threshold, we have the worst-case scenario of token misclassification. Nonetheless, this is not a problem in practice: if a filter cannot be tightly approximated by our distillation procedure, we can chose not to distill that specific filter and assign to it a state-dimension $d=L$. While we wouldn't get any speedup in that convolution, we can make sure we don't have any degradation in performance. It is also worth mentioning that in all the models we distilled, we have yet to experience the impossibility of reducing the approximation error sufficiently not to cause token misclassification with more than a few state dimensions.
>
> > Notwithstanding the theory of the tightness of that last approximation, there inevitably arises some situation where none of the approximations match the Transformer in quality.
>
> These are important points we are happy to elaborate upon. The proposed modal parametrization is capable of implementing associative recall over long sequences, even when trained from scratch (see results in general response and below). In [1] Hyena models have already been shown to match Transformer performance up to 350M scale. In this work, we investigate whether distillation of Hyena models into a recurrence is possible, without affecting downstream performance. Our perspective is that pretraining with implicit long convolutions and then distilling may be a more fruitful avenue that designing approximate or sparse variants of attention to train from scratch.
>
> > How well does the LaughingHyena architecture perform on the associative recall task, especially as compared to Hyena (or MultiHyena)? Is the point beyond which the models fail to perform the task (in terms of sequence length or vocabulary size) different between the two?
>
> We provide additional ablations on the *associative recall* task to investigate the performance difference on recurrent models trained from scratch. We note similar results of this type i.e. associative recall benchmarking with long convolutions (Hyena) and state-space models (H3), and finite-length filters (Conv1D), are also provided by [1].
>
> | **Model**      | **10, 2k** | **40, 2k** |**10, 64k** | **40, 64k** |
> | ----------- | ----------- |  ----------- | ----------- |----------- |
>   |MultiHyena | 100 | 100 | 98 | 99 |
>   |LaughingHyena-16 | 100 | 82 | 98 | 85 |
>   |LaughingHyena-8 | 100 | 86 | 99 | 87|
>   |LaughingHyena trunc | 81 | 48 | 21 |12 |
>   |MultiHyena -> LaughingHyena-8 | 100 | 100 | 98 | 99 |
>
> Where the first number indicates vocabulary size, the second one sequence length. Some observations on the above: (a) direct truncation performs much worse on associative recall, (b) pretraining with the proposed modal parametrization can solve challenging associative recall but is outperformed by MultiHyena, and (c) pretraining a MultiHyena and then distilling incurs in no accuracy cost.
>
> > For LM-Eval-Harness and HELM, have you tried a baseline of taking the impulse response from Hyena/MultiHyena and truncating it to a finite impulse response? A sliding window seems like one of the simplest approximations to try in the world of convolutions, and it would be helpful to know if maybe some defect of the tasks or the base model results in nothing more being required. This would, in fact, be a useful baseline to have in the paper.
>
> We reran benchmarks with the suggested FIR-truncation baseline. Please see the results below (and additional details in the general response)
>
> | **Model**      | **Hellaswag** | **PIQA** |**OpenbookQA** |
> | ----------- | ----------- |  ----------- | ----------- |
> | Pythia 410M      | 39.2       | 66.8       |25.2       |
> | LLaMA 360M   | 40.4        | 67.2      | 27.5      |
> | MultiHyena 410M  | 41.3        | 69.2       |27.4       |
> | LaughingHyena-16  | 41.4        | 69.1       |27.3       |
> | LaughingHyena-8   | 32.5        | 58.4       |24.8       |
> | LaughingHyena trunc.  | 0.0        | 50.4       |21.4       |
>
> [1] Hyena Hierarchy: Towards Larger Convolutional Language Models

---

> > ### Comment · Reviewer_6dqz · 2023-08-19
> >
> > Thank you for the update.
> >
> > The results on the associative recall task and the comparison to a truncation baseline have help convince me that the distillation approach maintains interesting model behavior beyond what could be achieved with simple context truncation. I would recommend including these results in future revisions of the paper (or supplemental).
> >
> > I've updated my score based on the author response.

---

> > > ### Author Response · Authors · 2023-08-20
> > > **Response**
> > >
> > > Thank you for the detailed review and for suggesting interesting ablations!

---

### Official Review · Reviewer_wEPT · 2023-07-09

**Soundness:** 3 good
**Presentation:** 3 good
**Contribution:** 3 good
**Rating:** 7
**Confidence:** 2

**Summary:**

This paper proposes LaughingHyena - an improvement to the Hyena model that can perform long-convolutions instead of attentions in transformers to avoid the quadratic scaling issues. One of the issues with the convolution sequence models is that they incur significant cost due to autoregressive inference. To avoid this, this paper seeks to come up with a techinique to have constant memory recurrent inference to increase generation thoroughput. This is achieved using the use of compact linear SSM and weight tying the filters across heads in Hyena architecture. The resulting performance improvements are impressive - 1.5x throughput improvement compared to Hyena. The model also achieves SOTA in the PILE dataset.

**Strengths:**

- The perplexity on small-scale models on Table 1 and Table 2 outperform GPT, Hyena and establishes a new SOTA.
- The peak memory usage is also constant for different sequence lengths in Table 5.4

**Weaknesses:**

- The writing is a bit hard to follow.
- The performance of the model is still lacking compared to full transformer baseline such as Pythia in Table 5.3. Can the authors comment on this? Any idea on how much the performance degradation will be on very large scale models and datasets?

**Questions:**

- What assumptions do you use for the state-space model in Eq 3.1 to yield a good distillation results (d<<L)?


**Limitations:**

I think the writing can be improved to provide a simple explanation of the method for readers who don't have a strong understanding of state-space models. Else, the text is hard to follow.

---

> ### Author Rebuttal · Authors · 2023-08-09
>
> We appreciate your thoughtful critique and questions and are happy to are happy to see positive feedback on the performance improvements.
>
> > Any idea on how much the performance degradation will be on very large scale models and datasets?
>
> It is certainly a fruitful to investigate whether distillation is still possible at larger scale. We pretrain larger models (~350M and 1.5B) and provide the results below. Please see the general response for more details.
>
> | **Model**      | **Hellaswag** | **PIQA** |**OpenbookQA** |
> | ----------- | ----------- |  ----------- | ----------- |
> | Pythia 410M      | 39.2       | 66.8       |25.2       |
> | LLaMA 360M   | 40.4        | 67.2      | 27.5      |
> | MultiHyena 410M  | 41.3        | 69.2       |27.4       |
> | LaughingHyena-16  | 41.4        | 69.1       |27.3       |
> | LaughingHyena-8   | 32.5        | 58.4       |24.8       |
> | LaughingHyena trunc.  | 0.0        | 50.4       |21.4       |
>
> > Writing is a bit hard to follow
>
> This is important feedback for us - we would be happy to address this. Section 2 was designed to be a self-contained introduction to long convolution and state-space models, and many of the technical details of Section 3 have been moved to the Appendix. We would be grateful if the reviewer could point out specific Sections that could use reorganization or simplification.
>
> > The performance of the model is still lacking a bit compared to Pythia
>
> The pertained model performs on par or better than Pythia on most of the HELM and LM-Harness tasks. Since the original pertaining runs at smaller scales, we have been able to optimize Hyena and MultiHyena parameters further, please see the results in the general response.
>
> > What assumptions do you use for the state-space model in Eq 3.1 to yield a good distillation results ($d << L$)?
>
> If you want to distill a length-$L$ ($L>> 1$) convolutional filter $h_t$ into a linear-time-invariant state-space model with a state dimension $d$ much smaller than $L$, you'd be looking to approximate the convolutional filter using a smaller, more compact representation. We know that with $d=L$ we can obtain a perfect state-space representation. When $d< L$, we are looking at representing the filter with
>
> $
>   \hat{h}_t = c^* A^{t-1}b,~~ B,C \in \mathbb{C}^d, A \in \mathbb{C}^{d \times d}
> $
>
> i.e. we are assuming that there exists a set of $d$ complex exponential basis functions which can approximate $h_t$.
>
> Several further practical considerations come into play:
>
> 1. *Smoothness of $h_t$*: If $h_t$ is smooth in $t$, and doesn't have high-frequency components, it is usually easier to approximate it with a lower-dimensional state-space model whose parameters are trained via gradient descent. The Hyena filters are the output of a shallow neural network with smooth nonlinearities (e.g. complex exponential positional encoding and sinusoidal activation).
>
> 3. *Low-rank Structure*: If the filter can be decomposed such that most of its energy / information is contained within a few modes or components (similar to principal component analysis), then a reduced-order approximation can capture most of the filter's behavior. This can be *numerically assessed on real filters* by inspecting the singular values of the Hankel operator constructed with the filter. The Ho-Kalman theorem [1] (reported in text as Theorem 3.1) establishes that the minimum dimension $d$ (the *Mc Millan degree*)  is equal to the rank of the Hankel operator while the AAK theory [2] leads to rigorous bounds on the best-case approximation error for a given filter and target state-dimension.
>
>    In our experiments, we show how the Hyena filters lose rank during pre-training and we can reliably use a state dimension $d << L$, predicted by the Hankel SVD, to distill the filters.
>
> 4. *Stability*: Hyena filters are implicitly regularized for recency bias by a learnable exponential window function: $h_t = e^{-a t} g(t)$. This practically leads the filters to be convergent for $L -> \infty$. In the context of distillation, this means that the state-space representation should be stable, i.e. that its poles should lie inside the unit circle. While we usually don't enforce it LRU-style [3] in our parametrization, we initialize the poles inside the unit circle.
>
> 6. *Error Tolerance*: Depending on the application, you'll need to decide how much error is acceptable between the convolutional filter $h_t$ and its state-space representation. A model with $d << L$ may not perfectly reproduce $h_t$ for all the Hyena filters, so you'll need to find a balance between approximation quality and dimensionality reduction. For example, we see that the filters in certain layers require higher state dimensions than others.
>
> 7. *Nature of Inputs*: If the filter $h_t$ is primarily used with a certain type or class of inputs, it's possible to optimize the state-space representation for that specific input. Knowing the typical input characteristics could help in achieving a good approximation with fewer states. However, our distillation methods want to be agnostic on the input characteristics.
>
> [1] Ho, B. L., and R. E. Kalman. "Effective construction of linear state-variable models from input/output functions." REGELUNGSTECHNIK (1966).
>
> [2] V.M. Adamjan, D.Z. Arov, M.C. Krein, "Analytic properties of Sc~nidt pairs for a Hankel operator and the generalized Schur-Takagi problem", Mat. USSR Sbornik, vol. 15, p. 31-73, 1971.
>
> [3] Orvieto, Antonio, et al. "Resurrecting recurrent neural networks for long sequences." arXiv preprint arXiv:2303.06349 (2023).

---

> > ### Author Response · Authors · 2023-08-20
> > **Response**
> >
> > Thank you for the comments and review! Since the discussion period is about to end, we leave a gentle reminder to take into account our response into the final evaluation of our work.

---

### Official Review · Reviewer_4kyN · 2023-08-04

**Soundness:** 4 excellent
**Presentation:** 3 good
**Contribution:** 3 good
**Rating:** 7
**Confidence:** 5

**Summary:**

This paper proposes an approach that enables constant-memory, recurrent inference for long convolution architectures. They introduce LaughingHyena, a distilation approach that consists of extracting compact linear SSMs from each convolution layer. Combined with weight-tying, it results in state-of-the-art performance and efficiency (i.e. throughput) without any drop in quality.

**Strengths:**

The paper is well structured and written.

The approach seems sound, reasonable and is performant

**Weaknesses:**

The models used in most experiments are small.

The helm evaluation is not very convincing.

**Questions:**

Is it possible to benchmark against more recent open source models such as Llamma?

**Limitations:**

Broader Impacts section is missing.

---

> ### Author Rebuttal · Authors · 2023-08-09
>
> We thank the reviewer for their comments and feedback.
>
> >  The models used in most experiments are small
>
> Our main objective with this work has been to verify whether it is possible to distill pertained long convolution language models into a recurrent form, and investigate the trade-offs induced by different distillation methods. However, we certainly agree that it is also valuable to evaluate whether scaling affects the optimal choices of distillation procedures. To check performance at larger scale, we pretrain larger Hyena models (**~350M** and **1.5B**, with **3B** and **7B** still in training) on The Pile, then distill following the procedure outlined in the paper (with the addition of a truncation baseline suggested by reviewer 6dqz). We observe no degradation with distillation into the proposed modal parametrization with at least 16 states. Please see the general response for additional details on these results.
>
> | **Model**      | **Hellaswag** | **PIQA** |**OpenbookQA** |
> | ----------- | ----------- |  ----------- | ----------- |
> | Pythia 410M      | 39.2       | 66.8       |25.2       |
> | LLaMA 360M   | 40.4        | 67.2      | 27.5      |
> | MultiHyena 410M  | 41.3        | 69.2       |27.4       |
> | LaughingHyena-16  | 41.4        | 69.1       |27.3       |
> | LaughingHyena-8   | 32.5        | 58.4       |24.8       |
> | LaughingHyena trunc.  | 0.0        | 50.4       |21.4       |
>
>  > The HELM evaluation is not very convincing.
>
> Thank you for the feedback, and are open to suggestions and clarifications on specific aspects of the evaluation. We highlight that the evaluation has been designed primarily to verify whether distillation without quality degradation is possible, and as a secondary goal to confirm that long convolution language models (e.g., Hyena) are competitive with Transformers across tasks. For the benchmarks in Table 5.3, we follow the standard HELM (or LM-Harness) evaluation procedure. We welcome further questions and would be happy to have the chance to further clarify with additional details.
>
>
> > Is it possible to benchmark against more recent open source models?
>
> Please see the general response for direct comparisons to a model using the LLaMA architecture.
>
> > Broader Impact is missing
>
> The Broader Impact section can be found in the supplementary material, below the table of contents.

---

> > ### Comment · Reviewer_4kyN · 2023-08-15
> > **Update**
> >
> > This response together with the general author response addresses my key questions. I have updated my score to reflect this.
> >
> > Please update this article with the new information and tables.

---

> > > ### Author Response · Authors · 2023-08-20
> > > **Response**
> > >
> > > We are glad to hear our response was useful! Thank you again for the review and engagement.

---

### Author Rebuttal · Authors · 2023-08-09

**General Response**

We thank all reviewers for their comments and valuable questions. In this work, we study long convolution language models (H3 [1], Hyena [2]) and propose a novel methodology to distill recurrences for fast inference at generation time. Insights on the structure of the learned convolutional filters further reveal improvements to the general Hyena architecture, validated empirically (Tables 5.1, 5.2) and theoretically (Theorem 4.1).

We are happy to have received positive feedback on **execution, experiments and empirical evaluation** ("*sound, reasonable and performant*" **4kyN**, "*impressive improvements*" **wEPT**), **novelty** ("*novel and meaningful*" **n6Pj**, "*novel method of linking convolutions and state-space models*" **6dqz**) and **theorical contributions** ("*theoretically grounded distillation*" **6dqz**).

In this response, we address the main comments related to presentation, technical details of distillation, and scaling.

**Distillation at larger scale**

Below, we are excited to share some additional experiments on larger models ("*[...] how much does performance degrade at a larger scale?*" **wEPT**, "*Is it possible to benchmark against more recent open source models*" **4kyN**). All models are trained on The Pile, for 300B tokens, same tokenizer, with LLaMA 360M trained from scratch using the LLaMA architecture:

| **Model**      | **Hellaswag** | **PIQA** |**OpenbookQA** |
| ----------- | ----------- |  ----------- | ----------- |
| Pythia 410M      | 39.2       | 66.8       |25.2       |
| LLaMA 360M   | 40.4        | 67.2      | 27.5      |
| MultiHyena 410M  | 41.3        | 69.2       |27.4       |
| LaughingHyena-16  | 41.4        | 69.1       |27.3       |
| LaughingHyena-8   | 32.5        | 58.4       |24.8       |
| LaughingHyena trunc.  | 0.0        | 50.4       |21.4       |


Direct truncation techniques for distillation ("*have you tried a baseline of taking the impulse response from Hyena/MultiHyena and truncating it to a finite impulse response?*" **6dqz**) do not preserve downstream performance, further motivating distilling onto a different basis. We truncate at 64 values, a 4x increase over the 16 states required by LaughingHyena-16. We also tested distillation on a 1.5B parameter MultiHyena trained for 180B tokens, and observe once again that 16 states are sufficient to preserve downstream performance.

**Direct associative recall with modal and truncated parametrizations**
Further, we provide additional ablations on the *associative recall* task ("How well does the LaughingHyena perform on the associative recall task", 6dqz), to investigate the performance difference on recurrent models trained from scratch. We note similar results of this type i.e. associative recall benchmarking with long convolutions (Hyena) and state-space models (H3), and finite-length filters (Conv1D), are also provided by [2].

| **Model**      | **10, 2k** | **40, 2k** |**10, 64k** | **40, 64k** |
| ----------- | ----------- |  ----------- | ----------- |----------- |
  |MultiHyena | 100 | 100 | 98 | 99 |
  |LaughingHyena-16 | 100 | 82 | 98 | 85 |
  |LaughingHyena-8 | 100 | 86 | 99 | 87|
  |LaughingHyena trunc | 81 | 48 | 21 |12 |
  |MultiHyena -> LaughingHyena-8 | 100 | 100 | 98 | 99 |

Where the first number indicates vocabulary size, the second one sequence length. Some observations on the above: (a) direct truncation performs much worse on associative recall, (b) pretraining with the proposed modal parametrization can solve challenging associative recall but is outperformed by MultiHyena, and (c) pretraining a MultiHyena and then distilling incurs in no accuracy cost.

**Effect of head dimension on pretraining perplexity**

Finally, we investigated the effect of the head dimension on Pile pretraining (results are improved over numbers presented in Table 5.2 due to further hyperparameter optimization).

| **Model (~155M)**      | **10B** |
| ----------- | ----------- |
| MultiHyena-4     | 10.8     |
| MultiHyena-8   | 10.4        |
| MultiHyena-16  | 10.1      |
| MultiHyena-32  | 9.6       |

[1] Hungry Hungry Hippos: Towards Language Modeling with State Space Models

[2] Hyena Hierarchy: Towards Larger Convolutional Language Models

---

### Decision · Program_Chairs · 2023-09-21

**Decision:**

Accept (poster)

**Comment:**

All reviewers find the paper interesting and has sufficient contributions extending long convolution based alternatives to attention. There were concerns around small scale and limited experiments and writing clarity. Authors response sufficiently addressed these concerns with additional experimental results. All reviewers agree for acceptance. One high level comment is about lack of discussion and comparisons with efficient attention approaches such as sparse attention and memorizing transformers. Currently the paper only focuses on comparison to similar approaches, and adding more discussion/comparison with other efficient attention approaches will make it easy for readers to better appreciate the results. Including results on more long context benchmarks like scrolls, can also help strengthen the paper.